

# Spatial variability of seasonal precipitation lapse rates in complex topographical regions - application in France

Valentin Dura[1,2], Guillaume Evin[2], Anne-Catherine Favre[3], and David Penot[1]

[1]EDF-DTG, Grenoble, France
[2]Univ. Grenoble Alpes, INRAE, CNRS, IRD, Grenoble INP, IGE, 38000 Grenoble, France
[3] Univ. Grenoble Alpes, CNRS, IRD, Grenoble INP, Grenoble, France

**Correspondence:** Valentin Dura (valentin.dura@edf.fr)

**Abstract.** Seasonal precipitation estimation in ungauged mountainous areas is essential for understanding and modeling a physical variable of interest in many environmental applications (hydrology, ecology, cryospheric studies). Precipitation Lapse Rates (PLRs), defined as the increasing or decreasing rate of precipitation amounts with the elevation, play a decisive role in high-altitude precipitation estimation. However, the documentation of PLR in mountainous regions remains weak even though their utilization in environmental applications is frequent. This article intends to assess the spatial variability and the spatial-scale dependence of seasonal PLRs in a varied and complex topography region. At the regional scale (10,000 km$^2$), seven different precipitation products are compared in their ability to reproduce the altitude dependence of the annual/seasonal precipitations of 1,836 stations located in France. The Convection-Permitting Regional Climate Model (CP-RCM) AROME is the best in this regard, despite severe precipitation overestimation in high altitudes. The fine resolution of AROME allows for a precise assessment of the influence of altitude on winter and summer precipitations on 23 massifs at the sub-regional scale ($\sim$ 1,000 km$^2$) and 2,748 small catchments ($\sim$ 100 km$^2$) through linear regressions. With AROME, PLRs are mostly higher in winter at the catchment scale. The variability of PLR is higher in high-altitude regions such as the French Alps, with higher PLRs at the border than inside the massifs. This study emphasizes the interest of conducting PLR investigation at a fine scale to reduce spatial heterogeneity in the seasonal precipitations–altitude relationships.

## 1 Introduction

Precipitation Lapse Rates (PLRs) refer to the increasing or decreasing rate of precipitation amounts with elevation. The rise of moist air above an orographic barrier such as hills or mountains generates precipitations that affect the ice cover and the precipitation amount contained in snowpacks (Bales et al., 2006; Viviroli et al., 2007; Mott et al., 2014; Dozier et al., 2016; Wrzesien et al., 2019). Therefore PLR is the main factor that controls the water budget of high-altitude catchments (Jiang, 2003) and partly explains the freshwater supply in the summer season. On shorter time scales like hourly or daily scales, PLRs help understand the physical mechanisms of orographic precipitations responsible for natural hazards such as avalanches, landslides, and floods (Caracena et al., 1979; Caine, 1980; Conway and Raymond, 1993; Buzzi et al., 1998; Panziera et al., 2015). PRLs can also be of importance for hydrological modeling, for which precipitation inputs are often spatialized with geostatistics models (Daly et al., 1994; Gottardi, 2009; Frei and Isotta, 2019) or can be distributed by altitudinal bands using assumed



PLR values (Bergström, 1992; Ragettli and Pellicciotti, 2012; Markstrom et al., 2015; Garavaglia et al., 2017; Ruelland, 2020; Kumar et al., 2022) to complete the hydrological balance (Oudin et al., 2006) in high-altitude catchments. The investigation of PLR is a topical and crucial issue in multiple domains such as energy production, agriculture, tourism, condition of ecosystems, risk management (Pimentel et al., 1997; Gössling et al., 2012).

The relationship between altitude and precipitation accumulation is not straightforward on small-time scales and depends
on precipitation type (stratiform *vs* convective), wind, and flow directions (Sevruk and Mieglitz, 2002; Schäppi, 2013). PLR at the hourly scale may even exhibit a negative trend during extreme events (Formetta, 2021) due to the drying of the air mass. When aggregated over extended periods, the relationship between precipitation and elevation becomes simpler to model, with generally a precipitation increase as elevation increases (Barrows, 1933; Spreen, 1947; Schermerhorn, 1967; Smith, 1979). However, PLR values can differ among seasons and regions due to different weather processes (Napoli et al., 2019; Ménégoz
et al., 2020). In the Alps, large-scale processes mainly generate winter precipitation. Convective processes are more frequent in summer.

In a hydrology framework, PLRs refers to the slope of the linear regression between precipitation and elevation on spatial areas (e.g. Sevruk, 1997; Gottardi, 2009; Ogrin and Kozamernik, 2020; Avanzi et al., 2021; Bell et al., 2022) rather than on vertical atmospheric columns. In some cases, the relationship in high-altitude regions appears to be non-linear with a threshold
impact (Schäppi, 2013; Napoli et al., 2019) due to the drying of the air masses, and the use of quadratic regression is then recommended (Mahmood et al., 2021).

PLRs are often estimated from ground measurements from rain gauges, observations of Snow Water Equivalent (SWE) (Avanzi et al., 2021) derived from manual coring or radiation, and snow/rain gauges totalizator (Gottardi, 2009) measures. Using only ground stations to estimate PLRs requires a sufficient coverage of high-elevation areas, which is rarely met in
practice (Hofstra et al., 2010). High-elevation stations are prone to precipitation under-catch (Groisman and Legates, 1994; Pollock et al., 2018) and snow-redistribution. Strong winds coupled with a significant amount of solid precipitation might even induce an annual underestimation of up to 25% (Sevruk, 1997). Moreover, the robustness of the regressions and the local influences of altitude on precipitation must be balanced because of the irregular spatial sampling of punctual observations (Gottardi, 2009). In practice, in sparse rain gauge regions, PLR can only be computed on vast areas and fails to reflect the local
relationship between altitude and precipitation.

As an alternative, gridded precipitation products can help analyze the spatial variability of PLR at a fine spatial scale. Radar products that indirectly measure precipitation from reflectivity have been extensively used in hydrology (e.g. Ochoa-Rodriguez et al., 2019). However, radar data are inherently biased in mountainous regions due to ground echoes and beam blockage (e.g. Berne and Krajewski, 2013), which results in an underestimation of PLR (Faure et al., 2019). Satellite precipitation products,
despite certain limitations, are another easily accessible source of data. The biases in mountain regions and the relatively low resolution of the data present challenges in calculating PLR from satellite data (Li et al., 2017). Composite products merging radar or satellite to rain gauge data (Champeaux et al., 2009; Nie et al., 2015; Nguyen et al., 2020) can also be an option for the assessment of PLRs. However, the latter products are sensible on the density of the rain gauge network and suffer where no rain gauge are available (Silverman et al., 2013; Shen et al., 2018; Frei and Isotta, 2019). Regional Climate models (RCM) have





been used in numerous studies to evaluate the variability of PLR at regional scales (e.g. Kotlarski et al., 2012; Cuo and Zhang, 2017; Ménégoz et al., 2020). All results show that PLRs obtained with RCM data are higher and more spatially varying than those using rain gauges. Convection-Permitting Regional Climate Models (CP-RCMs) have recently emerged as an appealing tool for producing fine-resolution ($1 - 4 \, \text{km}^2$) climate simulations (Rockel et al., 2008; Brousseau et al., 2016; Keuler et al., 2016; Belušić et al., 2020). CP-RCMs are a refined extension of RCM where deep convection parametrization, a dominant source of precipitation error (Hohenegger et al., 2008; Foley, 2010; Kendon et al., 2012), is turned off. To produce a long series of CP-RCM simulations, including precipitation, a Numerical Weather Prediction (NWP) model is often driven by a RCM or a satellite-based reanalysis. CP-RCMs produce better precipitation intensities and frequencies than RCMs during heavy events (Ban et al., 2021; Caillaud et al., 2021). CP-RCMs represent more accurately orographic precipitation than RCMs because they explicitly resolve deep convective processes and include a finer representation of the topography (Lucas-Picher et al., 2021). These improvements occur especially in summer when convective events play a major role (Ban et al., 2021). Today, CP-RCMs are considered more reliable than reanalysis (Lundquist et al., 2019) in ungauged mountainous areas, despite current precipitation overestimation (Gerber et al., 2018; Dallan et al., 2023).

This paper explores the spatial variability of winter and summer PLR across a complex topographical region, with a dense network of 1,836 stations used as ground truth. The first objective is to compare seven types of gridded products (interpolator, reanalysis, satellite, radar, CP-RCM) in their ability to capture the relationship between observed annual/seasonal precipitations and altitude. The second goal is the investigation of the spatial variability of winter and summer PLR values at the sub-regional ($\approx 1,000 \, \text{km}^2$) and catchment ($\approx 100 \, \text{km}^2$) scales using a gridded precipitation product without ground station assimilation. Nesting of catchments in sub-regions allows the study of the spatial-scale dependence of PLR to provide guidelines for future precipitation interpolation studies. To our knowledge, only Jiang et al. (2022) explores these questions in the Third Pole. In a sparse rain gauge region, the authors found an accurate PLR reproduction with downscaled ERA5 precipitation product (Jiang et al., 2021). They highlighted the spatial variability of PLR at the catchment scale and emphasized the importance of conducting PLR studies at the finest possible spatial scale.

This paper is organized as follows. Section 2 describes the study domain and the available precipitation products. Section 3 presents the method used to derive PLR. Section 4 shows the results of the reproduction and spatial variability of PLR by the different products considered. Section 5 compares the findings to the literature and discusses some perspectives. Section 6 gives the conclusions.

## 2 Domain and data under study

### 2.1 A complex mountainous region

Figure 1(**a**) shows the study domain, which corresponds to the intersection of the spatial extents of the seven gridded precipitation products. The topography of the region is complex and varied, including three major mountain chains: the Alps, the Pyrénées, and the Massif Central. The Massif Central is a mid-mountain range reaching 1,885 m in the center of the Southern half of France and covers 85,000 km², which makes it the largest mountainous region of the country (15 % of the total surface



area of France). The Massif Central is bordered by the Rhone Valley in its Western part and the Mediterranean coastal plain in its Southern part. This region is mainly composed of large plateaus ranging in altitude from 600 m to 900 m and enclosed by mountains with circular summits corresponding to extinct volcanoes. The altitude presents an asymmetrical profile with summits in the South and the East (Cévennes) and less elevated areas in the northwest. The landscapes are varied and contain limestone plateaus cut by deep canyons, mountain peaks, and deep river valleys. The Massif Central is exposed to the Western wind and captures the precipitation coming from the Atlantic Ocean.

The Alps are the highest mountain range in Europe, reaching 4,808 m at the Mont-Blanc. The chain has a 40,000 km$^2$ surface area and is 1,200 km long, crossing seven countries: France, Switzerland, Italy, Germany, Liechtenstein, Austria, and Slovenia. The French Alps are separated from the Massif Central by hills and are divided into the Northern and the Southern parts. The Northern Alps are higher and have large valleys. In the Southern Alps, valleys are narrower. The climate of the French Alps has three main influences. The moist air can come from the West Atlantic. The warm air comes from the Mediterranean and blows Northward. Mediterranean circulation from Italy phenomenon (Garavaglia et al., 2010) brings large amounts of snow in the most Easterly regions during the winter. The Northern Alps are more rainy than the Southern Alps, which have almost 300 days per year of sunshine. The French Alps is a territory of high altitude and complex topography. The significative variations in mountain elevation and exposure result in the coexistence of multiple climates at small spatial scales. Mountains create micro-climatic conditions and influence daily weather. The Jura is a sub-alpine mountain range located in the North of the French Alps. It covers a surface area of 5,000 km$^2$ and reaches 1,700 m. It is a mid-mountain range composed of plateaus with a mean altitude of 600 m.

The Pyrénées form a natural border between France and Spain by joining the Atlantic Ocean to the Mediterranean Sea over 430 km long. It covers 19,000 km$^2$ and reaches 3,200 m on the French side, making it the second-highest mountain range in France. Most valleys are running South-North. The Spanish part is more arid and rocky than the French part, which catches the precipitation from the Atlantic. The French Pyrénées splits into the Atlantic, Central, and Eastern parts. The relief rises steeply from sea level to hills in the Atlantic and Eastern parts. The Central Pyrénées host sharp peaks but are more rounded than the Alps because of erosion. The Atlantic part gets more precipitation because of rainy Northwest winds. The wind from the Southwest brings warmer air in summer. The topography is diverse, with snow-capped mountains, deep green valleys, and coastal hills. The remaining of France consists of plains and are not included in this study.

In this study, we consider three spatial scales: the regional scale (10,000 km$^2$), the sub-regional scale (1,000 km$^2$), and the catchment scale (100 km$^2$). PLR computation will be conducted at the sub-regional (massif) and catchment scales so it can have a hydrologic interpretation. In the French Alps, 23 sub-regions named "massifs" have been identified through climatological homogeneity of precipitation in Pahaut (1991). The surface areas of the massifs range from 450 to 1,600 km$^2$, with a median value of 870 km$^2$. The 2,748 catchments collected from Sandre (2020) have a mean surface area ranging from 0.009 to 1,000 km$^2$, with a median value of 100 km$^2$. Figure 1(**c**) illustrates the massifs and the catchments analyzed in this study. The altitude is represented with a digital terrain model of 1 km resolution. The study domain is divided into squares of 1 km$^2$ that are referred to as "pixels".



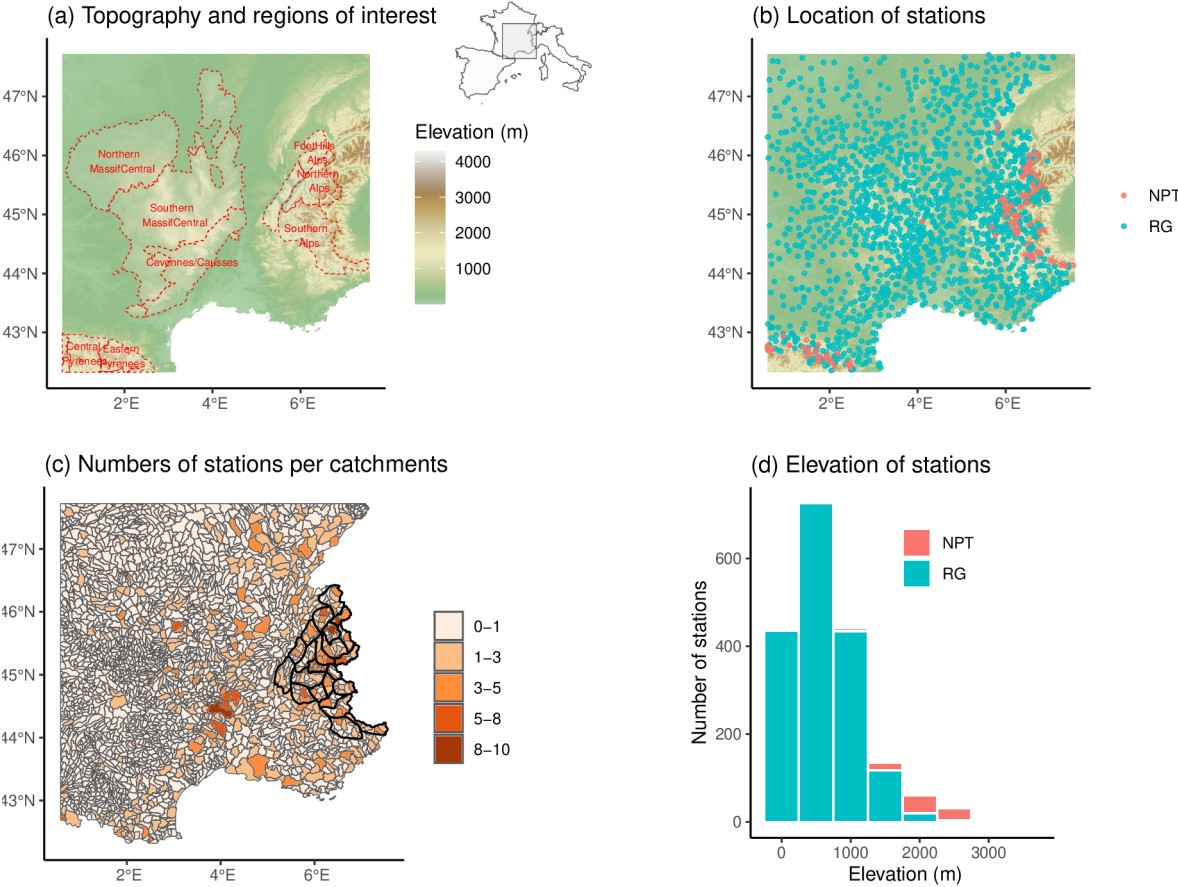

**Figure 1. (a)** Topography of France and boundaries of eight sub-regions of interest. The extent of the study domain is located in France and is bordered by Spain, Italy, and Switzerland. **(b)** Location of stations, blue points refer to rain gauges (RG) and red points to NPT (see subsection 2.2). **(c)** Catchments and the corresponding number of stations. Boundaries of SAFRAN massifs located in the Alps are also indicated in black. **(d)** Histogram of the altitude of stations (rain gauges, RG, and NPT).

## 2.2 A dense network of stations

Figure 1**(b)** presents the dense rain gauge network of 1,836 stations used in this study, with an average density of one rain gauge per 160 km$^2$. Electricité de France (EDF, the main electricity provider in France) and Meteo-France provide 673 and 1,163 stations, respectively. We aggregate available daily precipitation data to obtain seasonal and annual precipitation. Figure 1**(d)** shows the number of rain gauges per altitudinal band. A small number of rain gauges is available above 2,000 m, causing an under-representation of high-altitude regions. To overcome this limitation, EDF has implemented a snow rain gauges totalizator (Nivo Pluviomètre Totalisateur, NPT in French) device (Gottardi, 2009). NPTs collect annual precipitation in a small orifice protected from the wind by a large collar. Snow falling in the orifice feeds a reservoir containing a saline solution able to melt solid precipitation. The operating period ranges from 1945 to 1977 and differs among NPTs. We restricted the period to





1957–1973, where the spatial density of NPT is at its maximum. An exploratory analysis of the annual precipitation from NPTs removes outliers and missing values. Following this analysis, we keep 130 NPTs among 216 for the study. NPTs are primarily found in the Mont-Blanc region and are more uncommon, for instance, in the center of the Pyrénées. Figure 1**(c)** illustrates the irregular number of weather stations per catchment. Mountainous regions like the Alps and the Cévennes are more covered
140    than plains, where weather processes are less subject to spatial variability. Nevertheless, the Southern Alps, the Massif Central, and the Pyrénées have a complex topography and few stations. The sparse stations in some mountainous regions and their location in valleys largely constrain the ability to compute PLR using only precipitation aggregated from rain gauges.

In France, precipitation patterns differ among seasons. Thus in this study, four seasons are considered: winter (December, January, February (DJF)), spring (March, April, May (MAM)), summer (June, July, August (JJA)), and autumn (September,
145    October, November (SON)).

### 2.3    Gridded precipitation products

This study explores the ability of multiple gridded precipitation products to assess the relationship between annual precipitation and altitude. Precipitations from gridded products are available at daily, hourly, or sub-hourly timescales and aggregated to obtain seasonal precipitations. These products are briefly described hereafter and are summarized in Table 1.

150    #### 2.3.1    ERA5-Land

ERA5-Land (Muñoz-Sabater et al., 2021) is a global reanalysis that operates by simulating atmospheric and surface variables, including hourly precipitation. It uses the ERA5 atmospherical fields downscaled with linear interpolation at the resolution of 9 km. ERA5-Land assimilates satellite data and does not rely on rain gauges. The quality of ERA5-Land precipitation is dependent on the density of satellite and radar networks used in the assimilation process (Hassler and Lauer, 2021). As a
155    result, the quality is better in Central Europe and the U.S. than in tropical oceans. The improved horizontal resolution of 9 km compared to 31 km (ERA5) or 80 km (ERA-Interim) is a benefit of ERA5-Land (Muñoz-Sabater et al., 2021). This results in a better representation of spatial precipitation patterns (Gomis-Cebolla et al., 2023).

#### 2.3.2    CERRA-Land

Copernicus European Regional ReAnalysis Land (CERRA-Land) (Le Moigne, 2021) is the most up-to-date reanalysis available
160    in Europe since September 1984 at a horizontal resolution of 5.5 km. It uses ERA5 as lateral boundary conditions to run the atmospherical model HIRLAM ALADIN Regional/Mesoscale Operational NWP In Europe (HARMONIE) every 3 hours at a native resolution of 5.5 km. The analysis is given to the MESCAN system (Soci et al., 2016) to produce daily precipitation fields by merging it with nearly 8,000 daily rain gauge measurements through optimal interpolation. Le Moigne (2021) found that CERRA-Land leads to a better representation of the seasonality of snow depth in the Alps compared to ERA5-Land.



### 2.3.3 PDIR-NOW

PERSIANN Dynamic Infrared-Rain Rate (PDIR-NOW, Nguyen et al., 2020) is a satellite-based product using high-frequency (15 min) sampled infrared imagery, at a 4 km spatial resolution. PDIR-NOW estimates precipitation based on empirical cloud-top temperature–precipitation rate relationships. Errors resulting from this method are corrected by calibrating the empirical relationships regionally based on monthly precipitation climatology. PDIR-NOW has better diurnal cycle representation, rain/no rain days estimation, and regional precipitation patterns compared to the other PERSIANN family products (Nguyen et al., 2020). The inter-annual, annual, and seasonal precipitations are less biased with PDIR-NOW (Huang et al., 2021; Uysal, 2022). In the following, PDIR-NOW will be referred to as PDIR.

### 2.3.4 SERVAL

The *Système d'élaboration des produits radar et de visualisation centralisé* (SERVAL, previously PANTHERE, Champeaux et al., 2009) product provides precipitation amounts at a frequency of 5 minutes and a spatial resolution of 1 km by merging individual radar fields. Before any radars are combined, each one is individually calibrated using data from hourly rain gauges, with a mean hourly correction ratio applied uniformly across the whole radar coverage. If a pixel falls under the coverage of multiple individual radars, the selected precipitation corresponds to the maximum of precipitation seen by the considered radars. As already mentioned, radars are affected by many sources of potential errors, such as beam blocking and ground echoes. A precipitation probability product based on cloud classification filters clear air and ground echoes. Bright band and signal attenuation phenomena are also corrected.

### 2.3.5 COMEPHORE

The *COmbinaison en vue de la Meilleure Estimation de la Précipitation HOraiRE* (COMEPHORE, Champeaux et al., 2009) is a precipitation reanalysis based on the combination of the previous SERVAL product with a dense rain gauges data set of more than 4,000 observations. Kriging of the rain gauge values is applied to obtain the stratiform part, while the convective part involves the radar precipitation re-calibration with the rain gauge amounts. The main idea underlying this product is to combine the coherent spatial structure of the radars with the accuracy of rain gauge measurements to provide an hourly precipitation field at 1 km resolution. COMEPHORE underestimates mountains annual precipitation (Rouzeau, 2013; Roger, 2017) since the altitude is not considered at any modeling stages. Most radars have been integrated since 2006, and others have been gradually incorporated since 2015 to fill the gaps of measures in mountainous regions (Beck and Bousquet, 2013). The use of SERVAL and COMEPHORE is therefore tainted with temporal non-homogeneity.

### 2.3.6 SPAZM

The *SPAtialisation en Zones de Montagne* (SPAZM, Gottardi, 2009) is a precipitation interpolator that integrates rain gauge values with a meteorological guess conditioned by topography, and the weather type of the day. Eight weather situations are identified based on geopotential fields (Garavaglia et al., 2010) and eight corresponding mean daily precipitation fields



are created through local linear regression between mean daily observed precipitation at stations and altitude, at the daily timescale and 1 km resolution. For a given day, the corresponding mean daily precipitation field is modified using the precipitation of the current day. SPAZM incorporates 2,201 various stations from EDF (rain gauges and NPT), Meteo-France, MeteoSwiss, Arpa Piemonte, and Instituto Nacional de Meteorología (INM) networks. SPAZM used fewer but higher stations than COMEPHORE. Its ability to provide accurate estimates of yearly precipitation makes it a valuable tool for hydrological applications (Gottardi, 2009; Ménégoz et al., 2020; Ruelland, 2020).

### 2.3.7 CNRM-AROME

The CNRM-AROME model is a CP-RCM based on the nonhydrostatic, convective-scale, limited-area model called *Applications de la Recherche à l'Opérationnel à Méso-Echelle* (AROME) used for the national weather prediction by Meteo-France since 2008 (Seity et al., 2011; Brousseau et al., 2016). 37 years of CNRM-AROME simulations are available at a horizontal resolution of 2.5 km, at the hourly timescale (Caillaud et al., 2021), for an Alpine domain defined in the Flagship Pilot Study of the Coordinated Regional Climate Downscaling Experiment (CORDEX-FPS, Coppola et al., 2020). The Atlantic Pyrénées are not contained in the domain. The version of AROME applied to obtain this long simulation is related to cycle 41t1 (Termonia et al., 2018) and corresponds to the version in operational use at Meteo-France, between December 2015 and December 2017. In the remainder of the paper, CNRM-AROME CP-RCM will be referred to as AROME. The initial atmospheric conditions of AROME are provided every hour by a CNRM-ALADIN RCM simulation, itself driven by the ERA-Interim reanalysis. The fine resolution of the model enables a good representation of the deep convection scheme despite the absence of data assimilation. Precipitation from this version has been evaluated recently in multiple studies (Ban et al., 2021; Caillaud et al., 2021; Monteiro et al., 2022). Heavy precipitations are better represented than in the RCM CNRM-ALADIN but suspicious snow accumulation is pointed out in high altitudes. A first pre-processing stage is required to address large errors in precipitation seasonal amounts at the pixel scale. An iterative technique is applied to deal with the problematic pixels, whereby the abnormal values are replaced with the weighted (according to the altitude) average of the precipitation amount of the nearby pixels. A pixel is qualified as problematic if a difference of more than 500 mm in seasonal precipitations is noticed compared to a neighboring pixel.

**Table 1.** Summary of the characteristics of the gridded precipitation products used in this study, namely available period, domain, horizontal resolution, frequency, type of observations, provider, and reference.

| | Available period | Domain | Horizontal resolution | Frequency | Type of observations | Provider | Reference |
|---|---|---|---|---|---|---|---|
| ERA5-Land | 1950–2023 | World | 9 km | Hourly | Reanalysis ECMWF replay | ECMWF | Muñoz-Sabater et al. (2021) |
| CERRA-Land | 1984–2023 | Europe | 5.5 km | Daily | Reanalysis | C3S | Le Moigne (2021) |
| PDIR | 2000–2023 | 60°N to 60°S | 4 km | Hourly | Reanalysis Satellite | CHRS | Nguyen et al. (2020) |
| SERVAL | 2006–2020 | France | 1 km | 5min | Radar | Météo-France | Champeaux et al. (2009) |
| COMEPHORE | 2007–2020 | France | 1 km | Hourly | Reanalysis Rain gauges + radar | Météo-France | Champeaux et al. (2009) |
| SPAZM | 1950–2020 | France | 1 km | Daily | Reanalysis Rain gauges | EDF | Gottardi (2009) |
| AROME | 1982–2018 | Pan-Alps | 2.5 km | Hourly | CP-RCM | Météo-France | Caillaud et al. (2021) |



Figure 2 displays the mean annual precipitation for all gridded precipitation products. Fig. S1 in the Supplements provides the same information on a seasonal basis. We first notice the effect of the resolution. Annual precipitations from ERA5-Land and PDIR are very smooth on the considered domain compared to the other gridded products available at a finer resolution. The precipitation difference between plains and mountains is not very prominent in the radar products SERVAL and COMEPHORE. Comparatively, the Pyrénées, the Massif Central, and the Alps receive more annual precipitation with AROME, CERRA-Land, and SPAZM. SPAZM leads to marked patterns in the alpine valleys and produces a higher correlation to altitude. AROME gives a large amount of precipitation in mountains, especially in the Massif Central. The seven gridded precipitation products show different precipitation amounts, especially in mountainous regions, due to more or less fine horizontal resolutions, different types of assimilated data, and differences in the treatment of the altitude effect.

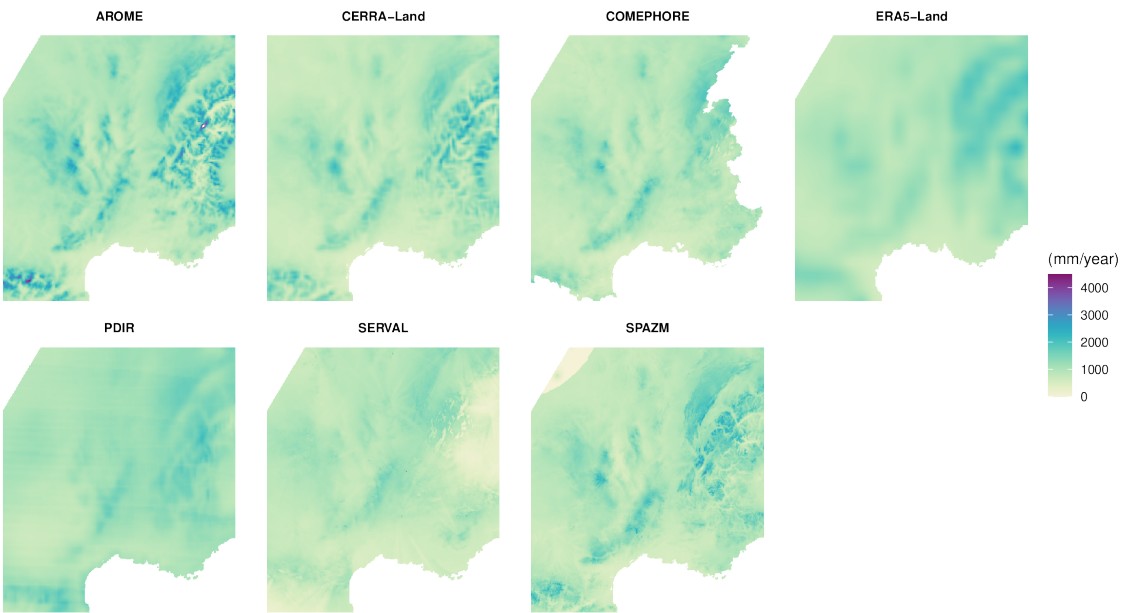

**Figure 2.** Mean annual precipitation for the gridded precipitation products (AROME, CERRA-Land, COMEPHORE, ERA5-Land, PDIR, SERVAL, SPAZM), on the period 2007–2017. An upper limit of 4,500 mm has been fixed for better visibility but values can be higher.

## 3  Methods

Let $Pobs$ and $Pgrid$ denote respectively the observed (by rain gauges and NPT) and gridded precipitation (annual or seasonal) expressed in mm. Let $Z$ indicate the altitude in m. The indices $i$ and $j$ refer to stations while the index $p$ indicates the pixels.





### 3.1 Description of the relationship between precipitation and altitude

Annual and seasonal precipitations from the gridded products are extracted at station locations to evaluate the reproduction of the altitudinal variability of precipitation. The precipitations from the NPT are only available at the annual scale. The stations

are grouped by regions defined in Fig.1(a) and by 300-meter elevation bands $b_{alt}$ ranging from 0–300 m to 3000–3300 m. For each elevation band and region, the median of observed ($Pobs_{b_{alt}}$) and gridded ($Pgrid_{b_{alt}}$) precipitation (annual and seasonal) is computed using all the locations of the stations belonging to this region and elevation band:

$$Pobs_{b_{alt}} = P\hat{obs}_{i_{50\%}}, i \in b_{alt}, \tag{1}$$

$$Pgrid_{b_{alt}} = P\hat{grid}_{i_{50\%}}, i \in b_{alt}. \tag{2}$$

The median values should represent the considered region and elevation band and be unaffected by anomalous values resulting from a specific station location. A compromise between a high number of rain gauges per band and a sufficient number of bands led to a 300 m bandwidth selection. The bands with fewer than five rain gauges are not displayed in Fig.3(a) to prevent any misunderstandings or misrepresentations. To study the behavior of gridded precipitation products in high altitudes where no rain gauges are available, we compare the median of seasonal precipitations $P\hat{grid}_{p_{50\%}}, p \in b_{alt}$ extracted from pixels within

the altitudinal bands.

To test a gridded precipitation product in reproducing the spatial variability of the relationship between seasonal precipitations and altitude at a finer scale, we also compute Precipitation Relative Difference (PRD) for several couples of close neighboring stations with sufficient elevation difference. The two stations should have a separation distance lower than 20 km and have an elevation difference superior to 100 m. PRD should not be assimilated to Precipitation Lapse Rates (PLR) calcu-

lated at the catchment scale, see subsection 3.2. For the neighbouring stations $i$ and $j$, we compute PRD at station locations for observed ($PRD_{Pobs_{ij}}$) and gridded ($PRD_{Pgrid_{ij}}$) precipitation :

$$PRD_{Pobs_{ij}} = 100 \times 100 \times \frac{Pobs_i - Pobs_j}{Z_i - Z_j} \bigg/ \frac{Pobs_i + Pobs_j}{2} \,, \tag{3}$$

$$PRD_{Pgrid_{ij}} = 100 \times 100 \times \frac{Pgrid_i - Pgrid_j}{Z_i - Z_j} \bigg/ \frac{Pgrid_i + Pgrid_j}{2} \,. \tag{4}$$

The gradient of seasonal precipitations relative to the altitude is divided by the mean precipitation of the two stations and multiplied one time by 100 to get the result in percentage relative to the mean precipitation. This result is again multiplied by 100 to express PRD in percentages per hundred meters (%/100 m).

### 3.2 Regression at nested spatial scales

The primary focus of this study is the investigation of the spatial variability of seasonal PLRs at the sub-regional and catchment

scales. It is possible to use stations at the regional scale, but the PLRs would be too dependent on their spatial sampling at the



sub-regional and catchment scales. Indeed, some catchments contain zero or only one station (Fig.1(**c**)). As an alternative, we propose to use AROME to study the spatial variability of PLR at the sub-regional and catchment scales. The choice of AROME will be motivated in subsection 4.1. We assume linear relationships between seasonal precipitations and altitude. This strong hypothesis will be discussed later in subsection 5.4. The PLRs are calculated with linear regression between seasonal
precipitations and altitude, expressed as follows:

$$Parome_p = \alpha + \beta \times Z_p + \epsilon_p, \text{with } \epsilon_p \sim \mathcal{N}(0, \sigma^2). \tag{5}$$

$Parome_p$ is the seasonal precipitations in mm for the pixel $p$, $\alpha$ is the y-intercept in mm, $\beta$ is the slope expressed in mm/m, and $Z_p$ is the altitude in m of the pixel $p$. The regression fit is performed using the maximum likelihood method implemented in the ℝ function *lm*. PLRs are derived from $\beta$ using the following formula:

$$PLR = 100 \times 100 \times \frac{\beta}{\bar{P}}. \tag{6}$$

The slope of the regression $\beta$ is multiplied by 100 twice, first to obtain the result as a percentage and once more to express PLR in percentage per hundred meters (%/100 m). The expression of PLR in %/100 m overcomes the biased nature of gridded precipitation products and allows a spatial comparison despite significant spatial and seasonal variation in precipitation amounts across France (Fig.2). To investigate the dependence of PLR on a given spatial scale, regressions will be conducted on 23
massifs of the Alps and catchments. On the catchments, a regression will only be performed if there is sufficient variability in elevation, *i.e.* if the standard deviation of altitude is higher than 50 mm.

## 4   Results

### 4.1   Ability of gridded precipitation products to reproduce the variability of the relationship between annual/seasonal precipitations and elevation

The altitude dependencies of the gridded precipitation products are compared to those from rain gauge data in the regions presented in Fig. 1(**a**). Figure 3(**a**) displays the median annual precipitation by 300 m altitudinal bands, for the seven gridded precipitation products, extracted at the location of the stations. In all regions of interest, annual precipitation observed at stations (*i.e.* the black curves showing the annual precipitations obtained with the rain gauges and NPT) increases as a function of the altitude. In the mid-range mountains such as the Northern Massif Central, the Southern Massif Central, and the foothills of
the Alps, the annual precipitation always increases from one altitudinal band to another. The maximum increase of observed annual precipitation occurs in the foothills of the Alps with a gain of 500 mm in 1500 m altitude. In more complex topographical regions such as the Northern Alps, the Southern Alps, and the Pyrénées, the enhancement of annual precipitation with the altitude is unclear, and only visible on a reduced range of altitude. For instance, the observed annual precipitation does not seem to increase in the Pyrénées below 1800 m. In mountain regions, even the mid-range ones like the foothills of the Alps,
ERA5-Land, PDIR, and SERVAL often lead to more moderate increases in annual precipitation with the altitude compared



to the stations. In the Cévennes/Causses region, annual precipitation from ERA5-Land slowly changes with altitude. In the Northern Massif Central, all gridded precipitation products, except for PDIR, are in agreement with the observed enhancement of precipitation with altitude. In the foothills of the Alps, the underestimation of SERVAL increases through the altitudinal bands, reaching 1000 mm at 1500 m altitude. AROME overestimates annual precipitation in all regions of interest, and the bias

generally increases at high altitudes. For example, at 2400–2700 m in the Southern Alps, AROME captures more than 1500 mm of median annual precipitation, while the stations record only 1000 mm. Nevertheless, median annual precipitation from AROME and observations are strongly correlated. Without data assimilation, AROME reproduces abrupt rises in precipitation at high altitudes. In the Northern Alps, AROME and the observed precipitations increase by the same amount of 500 mm from 1800–2100 m to 2100–2400 m. In some regions of interest such as the Pyrénées and the Northern Massif Central, the bias

of AROME is constant through the altitudinal bands. It reflects that, in those regions, the altitude impacts AROME and the station precipitations in the same way. The reanalysis CERRA-Land, based on rain gauges, is biased in some regions (Cévennes/Causses, Northern Alps, Pyrénées, Southern Massif Central). The bias is not constant and can be positive at low altitudes and negative at high altitudes such as in the Pyrénées region. The bias of CERRA-Land is relatively small in comparison to the bias of ERA5-Land, PDIR, SERVAL, and AROME as it reaches a maximum of 375 mm in the Northern Alps at 1800–2100 m.

The reanalysis COMEPHORE and the interpolator SPAZM accurately reproduce precipitation data from stations in all regions of interest, as both products assimilate rain gauge values. However, COMEPHORE underestimates precipitation above 2100 m, which is only visible in high-altitude regions, such as the Northern Alps and the Southern Alps. In the Northern Alps, the median annual precipitation from COMEPHORE is smaller than 1000 mm at 2400–2700 m, compared to more than 1300 mm measured with stations for the same altitudinal band. Figure 3(**a**) shows that COMEPHORE, CERRA-Land, and SPAZM

reproduce well the relationship between altitude and observed annual precipitation in the regions of interest. AROME, without rain gauge assimilation, exhibits the same precipitation/altitude trends that the stations, although it usually overestimates annual precipitation, especially at high altitudes. It can be added that the relationship between annual precipitation and altitude derived from the location of stations is dependent on the spatial sampling, which explains the variability of the curves in Fig.3(**a**). Nevertheless, the poor relationships between the annual precipitation obtained with ERA5-Land, PDIR, and SERVAL

and the altitude disqualify them for further analysis of the PRLs.

Figure 3(**b**) shows the median precipitations by altitudinal bands on a seasonal basis, at station locations (solid curves), and by considering all the pixels belonging to a region (dotted curves). The study of seasonal precipitations is extended above the altitude of the stations. The results are only displayed for the four highest altitude regions (Foothills of the Alps, Northern Alps, Pyrénées, Southern Alps). Only precipitation from AROME, CERRA-Land, COMEPHORE, and SPAZM are investigated

seasonally because of the highly biased nature of annual precipitation from ERA5-Land, PDIR, and SERVAL. Figure S2 in the Supplement presents the results for all seasons, products, and regions. At station locations (solid curves), the seasonal precipitations from AROME, CERRA-Land, COMEPHORE, and SPAZM are close to the observed ones. Only AROME overestimates seasonal precipitations, but the bias is not altitude-related. The relationship between seasonal precipitations and altitude is different at station locations (solid curves) and with all pixels (dotted curves) in the Northern and Southern Alps. The

dotted curves are more regular and above the solid ones, illustrating that the stations are not representative of the altitudinal





bands of the regions. When all pixels belonging to a region (dotted curves) are taken into account, differences grow among seasonal precipitations from AROME, CERRA-Land, COMEPHORE, and SPAZM. COMEPHORE experiences both lower winter and summer precipitation than AROME, CERRA-Land, and SPAZM. COMEPHORE precipitation remains unchanged or even decreases with altitude. As a result, the differences in precipitation amounts between COMEPHORE and the other

products become larger in high altitudes. For example, in the Northern Alps, the winter precipitations from AROME, CERRA-Land, and SPAZM increase by 250 mm between 3000–3300 m to 3300–3600 m whereas COMEPHORE declines between these two altitudinal bands, leading to a 350 mm gap. CERRA-Land and SPAZM show common seasonal precipitation patterns with altitude in the Northern Alps and the Pyrénées. In the Northern Alps, CERRA-Land and SPAZM agree on slow changes in seasonal precipitations below 3000 m and more rapid increases above this threshold. In the Southern Alps, the increase of

seasonal precipitations with altitude is larger with SPAZM than CERRA-Land. AROME produces a larger amount of seasonal precipitations in comparison to COMEPHORE, CERRA-Land, and SPAZM, especially in winter. In summer, AROME shows a suspicious precipitation amount of 1500 mm at high altitudes in the Northern Alps. In winter, the difference between AROME and the other gridded products is larger at high altitudes. For instance, in the foothills of the Alps, AROME matches SPAZM producing 250 mm at 0–300 m but is 300 mm higher at 2400–2700 m.

To investigate the ability of AROME, CERRA-Land, COMEPHORE, and SPAZM to reproduce the observed precipitation change with altitude at a fine spatial scale, Fig. 4 shows scatter plots of observed versus gridded precipitation relative difference (PRD). PRDs are defined for two neighboring (closest) stations as the difference in precipitation relative to the difference in altitude, see Sect. 3.2. PRDs are computed on all pairs of closest stations within the region of interest. Fig. S3 in the Supplement presents the PRDs scatter plots for all seasons. The same results observed in Fig. 3(**b**) appear. PRDs from COMEPHORE and

SPAZM are unbiased, the points being close to the 1:1 lines. The variance of PRD errors is higher with AROME and CERRA-Land. PRDs from CERRA-Land seem unbiased on average, and those from AROME are a little too high in winter. Fig. 4 also reveals more difficulties for AROME, CERRA-Land, COMEPHORE, and SPAZM to reproduce observed summer than winter PRDs. PRDs with AROME are unbiased on average. This is possible to use AROME to model the altitude effect on seasonal precipitations at a fine spatial scale.

Figure 5 illustrates the relationship between seasonal precipitations and elevation through two transects in the Cévennes and the Northern Alps. A transect indicates a virtual line crossing the topography. Figure S4 in the Supplements shows the transects for all seasons. Seasonal precipitations from SPAZM are, by construction, fully correlated to the altitude, reacting to slight variations of terrain. COMEPHORE and CERRA-Land are both less correlated to the altitude than SPAZM. CERRA-Land gives the same winter precipitation as SPAZM but different summer precipitation patterns. COMEPHORE produces

similar amounts of precipitation as SPAZM in the Cévennes transect but lower ones in the Northern Alps transect. AROME has a strong relationship with the altitude. It is the gridded product with usually the highest seasonal amounts in high altitudes. For example, winter precipitations reach 600 mm in the Cévennes with AROME, almost 500 mm with SPAZM, and 400 mm with COMEPHORE and CERRA-Land.

COMEPHORE and CERRA-Land sometimes exhibit a negative correlation to the altitude. The behavior of COMEPHORE

is surprising in the Northern Alps between 8 and 15 km after the beginning of the transect. COMEPHORE summer precipitation



**Figure 3.** Comparison between the altitude dependence of precipitation from AROME, CERRA-Land, COMEPHORE, ERA5-Land, PDIR, SERVAL, SPAZM, stations (rain gauges and NPT) in seven regions. The lines show the median precipitation amount in each altitude zone and the bars denote the number of rain gauges in each area. Comparisons are done at **(a)** the station locations at the annual scale and **(b)** both station locations and on all pixels from the gridded products at the seasonal scale. At the seasonal scale, only AROME, CERRA-Land, COMEPHORE, and SPAZM are analyzed for the sake of visualization. The season *djf* refers to the winter (December, January, February) and the season *jja* corresponds to the summer (June, July, August).




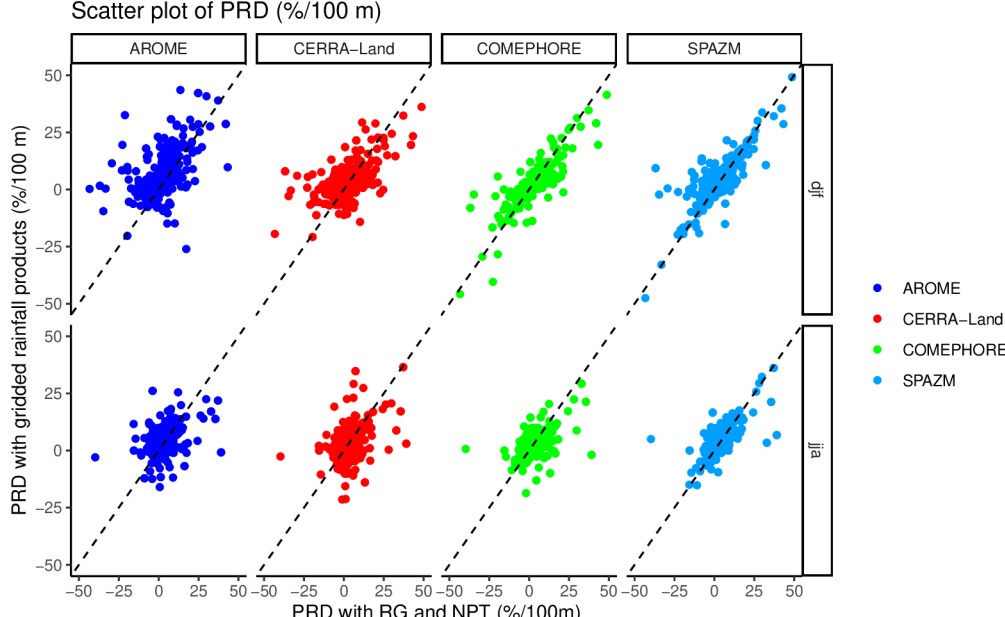

**Figure 4.** Comparison of precipitation relative difference (%/100 m) computed with stations and gridded rainfall products. PRDs are derived for all couples of neighboring stations. Scatter plot between observed and gridded PRD at station locations. The dotted lines indicate the 1:1 lines.

amount is the lowest for the highest pixel on the transect. CERRA-Land is different from the other gridded products in summer on the Cévennes with a negative association with the altitude for the first 80 km of the transect. SPAZM exhibits a higher correlation than AROME to the local topography. As a result, SPAZM precipitation maximums coincide with the summits while those of AROME are reached before the summits in the Northern Alps, for example. Another major difference between

AROME and SPAZM is the seasonality of precipitation. In the first 20 km of the Northern Alps transect, while SPAZM shows similar seasonal precipitation patterns, AROME seasonality is more marked with two peaks in precipitation in winter at 5 and 15 km and only one in summer. Moreover, in the Northern Alps transect, SPAZM produces higher summer precipitation than winter precipitation (400 mm against 300 mm), unlike AROME.

Overall, AROME and SPAZM demonstrate the ability to replicate the altitude-dependent variations of annual and seasonal

precipitations in the regions of interest. In comparison to commonly used gridded precipitation products such as satellite and radar data, AROME and SPAZM exhibit stronger correlations with observed precipitation. The evolution of AROME seasonal precipitations with altitude is consistent with that of stations, despite suspicious summer precipitation in very high altitudes in the Northern Alps and a too rapid rise of winter precipitation. In regions with sparse networks of stations, products based on stations such as SPAZM may be unable to capture the impact of local topography on precipitation. Since our objective is to

investigate the spatial variability of PLR at the catchment scale, SPAZM will be inadequate for this purpose. AROME should





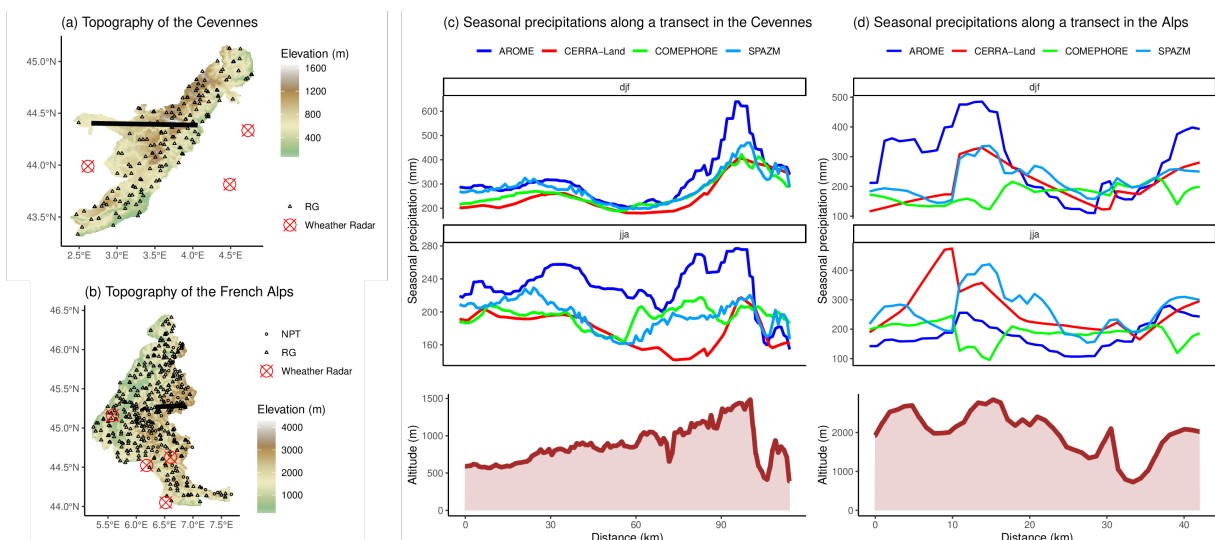

**Figure 5.** Transects in the Cévennes **(a)** and the Northern Alps **(b)**. Winter *djf* and summer *jja* precipitations extracted across the transects are colored according to gridded products **(c), (d)**. Elevation is represented in brown as an altitude profile.

be preferred to SPAZM in ungauged areas. PLRs are therefore computed with AROME simulations from 1982 to 2018. The long temporal period of more than 30 years allows for the investigation of PLR at the climatological scale.

## 4.2 Spatial and seasonal variability of Precipitation Lapse Rates

Initially, seasonal PLRs are computed at the sub-regional scale on the 23 massifs in the French Alps. Secondly, seasonal PLRs

are computed on catchments that form a subdivision of the massifs. Figure 6**(a)** displays the $R^2$ at the sub-regional (massif) scale of the regressions between AROME seasonal precipitations and the altitude. In winter, values of $R^2$ are close to 0.5 in the foothills of the Alps (Chartreuse, Bauges, Aravis, Chablais), in the Northern Alps (Beaufortin, Belledonne, Mont-Blanc), and in the Southern Alps (Queyras, Thabor, Oisans, Devoluy, Champseur, Pelvoux, Ubaye, Haut-Var-Haut-Verdon). Regression quality in the Central Alps (Grandes-Rousses, Maurienne, Vanoise, Haute-Tarentaise) is not high enough to conclude on PLR

values. Winter precipitation from AROME is for instance not correlated to the altitude in the Haute-Tarentaise massif ($R^2$ equals 0.12). In summer, the same spatial pattern is observed with lower values of $R^2$, indicating that the altitude is better correlated with winter precipitations than summer precipitations in the French Alps.

  The fine resolution of AROME provides the possibility to investigate the spatial variability of the relationship between seasonal precipitations and altitude at the catchment scale. Figure 6**(c)** shows the $R^2$ values at the catchment scale. $R^2$ obtained

from regressions are higher at the catchment scale than at the sub-regional scale for both winter and summer seasons. Similarly to the sub-regional scale, $R^2$s are generally higher in winter than in summer. In winter, regression quality does not show a clear spatial pattern except for lower $R^2$ in the Eastern Alps (Haute-Tarentaise, Haute-Maurienne, Thabor, Queyras). The $R^2$s show a large spatial variability. In the foothills of the Alps (Vercors, Chartreuse, Bauges, Aravis), strong $R^2$ (0.75–1) can be




found at the borders of the massifs next to small $R^2$ (0–0.5) inside the massifs. In summer, the regressions are of poor quality
in the Northern and Central Alps. The massifs in the Southern Alps and some in the foothills of the Alps (Chartreuse, Bauges,
Chablais) do not show a clear deterioration of $R^2$ compared to the winter. In some massifs like the Maurienne, in winter, the
$R^2$ are superior to 0.5 at the catchment scale and close to 0 at the sub-regional scale. In the same way, the massifs, where the
regression fits are correct at the sub-regional scale (Thabor, Haute-Maurienne), do not necessarily contain the best fit at the
catchment scale. It indicates that the previous regressions (Fig.6(a)) are likely conducted over excessively broad areas.

Figures 6(b),(d) respectively display the PLRs of AROME seasonal precipitations at the sub-regional and catchment scales.
We remark higher PLRs at the catchment scale (a mean of 5.43 %/100 m in winter and 3.31 %/100 m in summer) than at the
sub-regional scale (a mean of 4.73 %/100 m in winter and 2.79 %/100 m in summer). We consider that a $R^2$ inferior to 0.5
is too small to interpret the slope of the regressions. The spatial variability of seasonal PLRs can therefore not be investigated
at the sub-regional scale. We will only describe PLR values at the catchment scale because of better regressions. Figure 6(d)
reveals higher PLRs in winter with very high values (8-15 %/100 m) and high values (5-8 %/100 m) mainly located at the
border of the Alps (Devoluy, Thabor, Chablais). Moderate (3-5 %/100 m) PLRs are present in almost all massifs and do not
show a clear spatial pattern, such as a North-South separation. Small (0-3 %/100 m) PLRs are mainly found inside the massifs
of the foothills of the Alps (Vercors, Chartreuse, Aravis) and in the Far-Southern Alps (Mercantour, Haut-Var-Haut-Verdon).
In summer, the majority of PLRs are small (0-3 %/100 m), but some catchments at the border of the Alps (Bauges, Devoluy,
Haut-Var-Haut-Verdon) have moderate (3-5 %/100 m) or high PLRs (5-8 %/100 m). Figure 6 shows that the quality and slope
of the regression can vary widely across the French Alps, with higher values at the borders. A sub-region like the Vercors
massif covers around 1,350 km$^2$ and presents a considerable geographic heterogeneity in PLR values. Figures S5, S6, S7, and
S8 in the Supplements illustrate the results of the regressions for all seasons.

Figure 7 shows the winter and summer PLRs at the catchment scale in the study domain. Results for the other seasons are
available in Fig. S9 and S10 in the Supplements. The catchments located in the Morvan, the Massif Central, the Jura, the
Eastern Pyrénées, and the catchments close to the Mediterranean with enough altitude variation (standard deviation of the
altitude higher than 50 m) have the strongest PLRs. These values are higher than those of the Alps, as they can reach 32 %/100
m. Values between 8 and 15 %/100 m are also considerably more frequent than in the Alps. In the study domain, 40 % of
the catchments with $R^2 > 0.5$ have winter PLR greater than 8 %/100 m. This percentage drops to 15 % if we only consider
the Alps. The slopes of the regression have higher variability in the Alps and the Pyrénées. PLRs are seasonally varying, with
generally the highest values in winter. In winter, the Massif Central hosts the strongest PLRs. In summer, the Mediterranean
area, ranging from the Eastern Pyrénées to the Mediterranean Alps, has the highest PLRs.



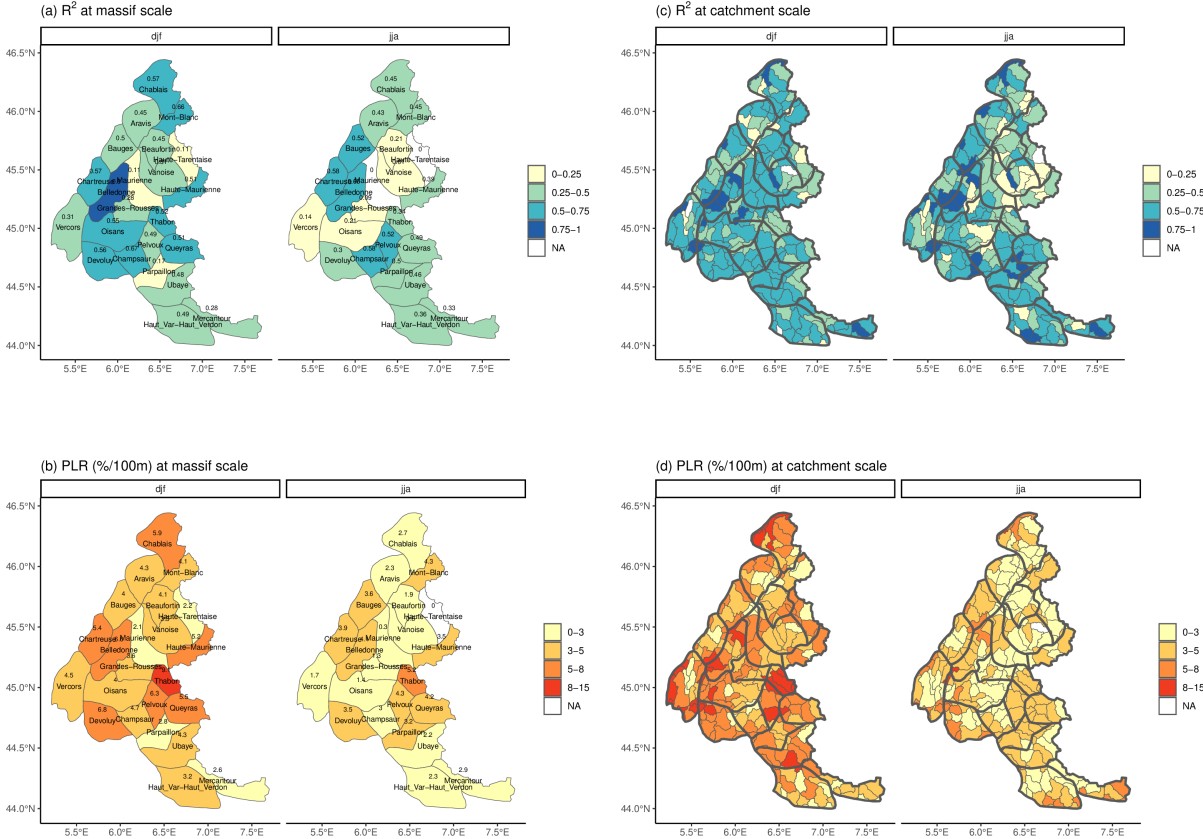

**Figure 6.** Spatial distribution of $R^2$ **(a), (c)** and precipitation lapse rates (precipitation change per 100 m altitude difference divided by area-average precipitation) **(b), (d)** extracted from the precipitation-altitude regressions for all grids within each area. Categorical results are displayed in the French Alps at regional and catchment scales for both winter *djf* and summer *jja* precipitations. The area indicated as "NA" and filled in white represents regions with insufficient altitudinal variability to compute precipitation gradient. The names of the massifs and their corresponding $R^2$ and precipitation gradient values are also printed.

## 5 Discussion

### 5.1 Ability of gridded precipitation products to capture the relationship between annual/seasonal precipitations and altitude


This article reviews the potential of seven gridded precipitation products for precipitation gradient estimation in a complex topographic region. ERA5-Land, PDIR, and SERVAL underestimate annual precipitation gradients even for mid-range mountains. The limitations of these three gridded precipitation products are even more apparent in high-altitude regions such as the



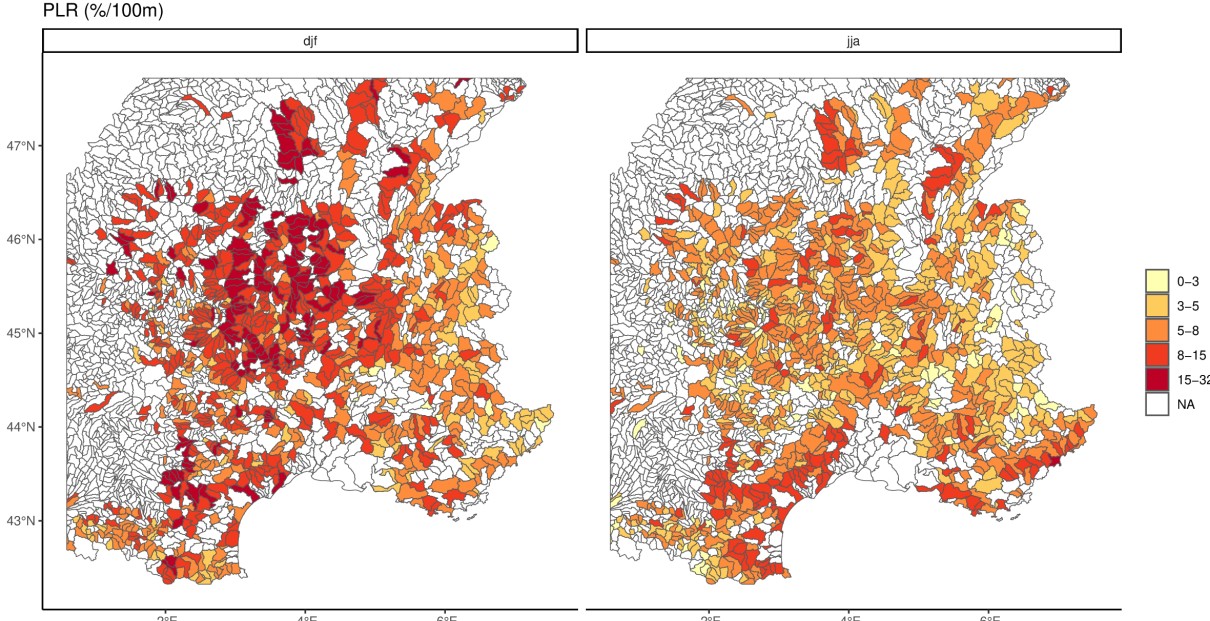

**Figure 7.** Spatial distribution of winter *djf* and summer *jja* precipitation lapse rates (precipitation change per 100 m altitude difference divided by area-averaged precipitation) extracted from the precipitation-altitude regressions for all grids within each area. Categorical results are displayed in the study domain at the catchment scale. The area indicated as "NA" and filled in white represents catchments with insufficient altitudinal variability (standard deviation of the altitude higher than 50 m) or $R^2$ inferior to 0.5.

Northern Alps and the Southern Alps. The assimilation of satellite data and the non-use of rain gauge values in ERA5-Land
and PDIR is an obstacle to their utilization in a complex topographic region such as France. These findings converge with those
of Jiang et al. (2022), where they found that the satellite product IMREG cannot reproduce the precipitation gradient in Tibet.
SERVAL does not use altitude as a co-variable to determine precipitation from individual radars. The uniform ratio applied for
this product in the hourly correction only represents the station elevation range, not the entire radar coverage. The translation
of reflectivity to precipitation amount using the vertical profile of reflectivity is another potential source of error that results
in inaccurate negative precipitation gradients in Fig.3. In their study of the French Alps at the sub-regional level, Faure et al.
(2019) found a significant underestimation of precipitation gradient using PANTHERE, SERVAL's predecessor.

CERRA-Land, COMEPHORE, and SPAZM assimilate ground precipitation measurements from weather stations and are
thus correlated with the annual precipitation observed at the stations. COMEPHORE leads to an adequate relationship between
annual/seasonal precipitations and altitude in mid-mountain ranges highly instrumented as the Southern Massif Central. How-
ever, in more complex topographical regions with sparse stations such as the Northern Alps, COMEPHORE relies heavily on
radar data from high altitudes and is unable to accurately reproduce the increase of precipitation with altitude. COMEPHORE



processes fewer high-altitude stations than SPAZM, making it unable to represent precipitation in high-altitude areas correctly. The sharp decrease of summer precipitation with altitude from COMEPHORE in the Northern Alps illustrated in Fig.5(**d**) is likely due to the shielding of the radar beams by mountain ranges (Germann et al., 2006). CERRA-Land can replicate the
observed seasonal precipitation enhancement with the altitude. However, at the annual scale, CERRA-Land is slightly biased in some regions, and the bias may change of sign according to altitudinal bands such as in the Pyrénées (Fig.3(**a**)). SPAZM can reproduce the relationship between annual precipitation at station locations and altitude at the regional scale. This result is not surprising as SPAZM interpolates precipitation observations from the stations, incorporating the effect of the altitude through local linear regressions. However, SPAZM is affected by the density of stations. In regions where the stations are more sparse,
as in the Pyrénées, annual precipitations from SPAZM do not match annual precipitations from the stations. This suggests a possible limitation of SPAZM in ungauged mountainous regions. AROME, without rain gauge assimilation, can reproduce the relationship between seasonal precipitations and altitude, despite a positive bias in winter in mountainous regions. We find that AROME produces higher annual/seasonal precipitations than the other gridded products. The difference in seasonal precipitations is limited at low altitudes and rises sharply at high altitudes. Higher precipitation accumulations with RCMs have been
documented. In Western Montana, in high-altitude where observations are sparse, Silverman et al. (2013) remarked higher annual precipitation with the Weather Research and Forecasting (WRF) model than with the Parameter-elevation Regressions on Independent Slopes Model (PRISM) (Daly et al., 1994). In India, Li et al. (2017) found that annual precipitations obtained with WRF are twice or triple the amount of the satellite precipitation product Tropical Rainfall Measuring Mission (TRMM). The correlation between AROME and COMEPHORE on the Cévennes transect in summer (Fig.5(**c**)) is comforting as the radars
operate well in summer in the Cévennes region (Delrieu et al., 2013). Moreover, the seasonality of precipitation is taken into account with AROME (see Fig.5(**c**)). AROME simulates precipitation without using rain gauge data. As a result, precipitation under-catch in high-elevation stations does not affect AROME simulations. Precipitation under-catch is more common in winter. This partly explains the precipitation difference between AROME and the other products in winter and the agreement in summer in Fig. 3.

**5.2 Values and spatial variability of Precipitation Lapse Rates**

Table 2 synthesizes the main research on PLRs. Like Ménégoz et al. (2020) in Switzerland, we find higher PLRs in winter, likely due to the large-scale circulation of air masses. In winter, the Western wind coming from the Atlantic loaded with moisture is dominant and hits the primary mountainous regions on its way. The affected catchments correspond to the first orographical barrier and are located in the Massif Central, extending from the Languedoc to the Morvan regions. On those
catchments, 50 % of the PLRs are in the range of 5–13 %/100 m. For the same reason, the PLRs are in the range of 5–15 %/100 for the first Western catchments of the Alps. Interestingly, a distinct pattern emerges in the Thabor where some PLRs are in the range of 8–15 %/100 m due to frequent Easterly weather fronts brought by Mediterranean circulation from Italy. In summer, the Mediterranean area is subjected to the dominant Southern flows and hosts the highest PLRs. The Southern air masses absorb the humidity from the Mediterranean Sea. For this reason, we found higher PLRs in the Eastern Pyrénées (8–15
%/100 m) compared to the Central Pyrénées (3–8 %/100 m). Convective processes are more frequent in summer than in winter.





Daily precipitation amounts from convective processes are often negatively correlated to the altitude (Schäppi, 2013). That is likely to explain the lower PLRs in summer than in winter. To resume, we notice high PLRs in areas subjected to the prevailing winds in France, which are Western, Southern, and Eastern winds depending on the season. Like Kotlarski et al. (2012), we find lower PLRs in the Alps than in the rest of France.

In the French Alps, a complex topographical region, we do not find a significant difference in PLR values between the Northern and the Southern Alps for the winter and summer seasons contrary to Sevruk (1997); Durand et al. (2009); Ménégoz et al. (2020). That is probably due to the finer spatial scale used in the study. At a larger spatial scale, large-scale circulation, rather than the local topography, drives seasonal precipitation amounts (Jiang et al., 2022). We find a large spatial variability of PLRs, which can vary inside the same massif from one catchment to another. For instance, winter PLRs range from 3–5

%/100 m to 8–15 %/100 m in Belledonne. Using rain gauges, Ogrin and Kozamernik (2018) highlighted the extreme spatial variability of PLRs in the range of 5–16 %/100 m for the summer season in three nearby alpine valleys of Slovenia. We find the same magnitude in PLR values and variability. PLRs are highly spatially varying in high-altitude and complex topographical regions such as the French Alps and Pyrénées. In the Massif Central, PLRs exhibit less spatial variability.

| Study domain | Data | Spatial scale | Period | Findings | Reference |
|---|---|---|---|---|---|
| Switzerland | Rain gauges | massifs 4,000 km$^2$ | 1951–1980 | Northern Alps: 5.4 %/100 m Southern Alps: 0.24 %/100 m | Sevruk (1997) |
| French Alps | SAFRAN | regions 1,000–10,000 km$^2$ | 1958–2002 | Northern Alps: 29.4 mm/100 m Central Alps: 19.5 mm/100 m Southern Alps: 17.2 mm/100 m Mediterranean Alps : 17.8 mm/100 m | Durand et al. (2009) |
| Europe | RCM | country 200,000–1,000,000 km$^2$ | 1961–2000 | Spain: -3.7 mm/100 m United-Kingdom: 131 mm/100 m France (without the Alps and the Pyrénées): 32 mm/100 m Alps: 14.6 mm/100 m | Kotlarski et al. (2012) |
| Slovenia | Rain gauges | | 2012–2015 (MJJASON) | 5–16 %/100 m | Ogrin and Kozamernik (2018) |
| Swiss and French Alps | RCM | regions 10,000–50,000 km$^2$ | 1971–2008 | French Northern Alps: 2.6 %/100 m French Southern Alps: 1.6 %/100 m Swiss Alps: 4.3 %/100 m in winter (DJF) 2 %/100 m in summer (JJA) | Ménégoz et al. (2020) |
| Troisième pôle | WRF | regions (1,000,000 km$^2$) catchments (10,000 km$^2$) | 1980-2018 | regions: 2.90–11.26 %/100 m Large catchments: -10–10 %/100 m Small catchments: -4–10 %/100 m | Jiang et al. (2022) |

**Table 2.** Summary of the studies on precipitation lapse rates (PLRs), namely study domain, data, spatial scale, period, findings, and reference.

## 5.3   Dependence of the Precipitation Lapse Rates to the spatial scale

The low $R^2$ found in Fig.6(a) is a reminder of the high intra-massif variability of the relationship between seasonal precipitations and altitude in large sub-regions. The sub-regions dissection into more homogeneous areas, such as small catchments with a surface area close to 100 km$^2$ allows a better description of PLRs. Regressions at the catchment scale have larger PLR and $R^2$ values than at the sub-regional scale. In winter in Maurienne, all catchments have $R^2$ greater than 0.5 with PLRs in the range 3–8 %/100 m. The $R^2$ is close to 0, and the slope is close to 2 %/100 m with a single regression covering all the



Maurienne massif. Figure 8 shows the scatter plot between AROME annual precipitation and altitude for the pixels within three groups of catchments in the Vercors, a massif covering 1,350 km$^2$. The colors of pixels relate to distinct geographical locations. Figure S11 in the Supplements provides the scatter plots for seasonal precipitations. Figure 8 suggests that the Vercors encompasses several sub-regions with different relationships between annual precipitation and altitude. This relationship is linear for the Southern and Western Vercors catchments. However, the Western and Southern Vercors are not subject to the same meteorological influences. The PLRs are higher in the Western Vercors because of the dominant rainy westerly flow. The relationship is not linear for the catchments located in the Vercors plateaus. The attenuation of orographic precipitation after crossing the first hills of the West of the massif is a possible explanation. Figure 8 reveals the coexistence of PLR spatial patterns within a single massif.

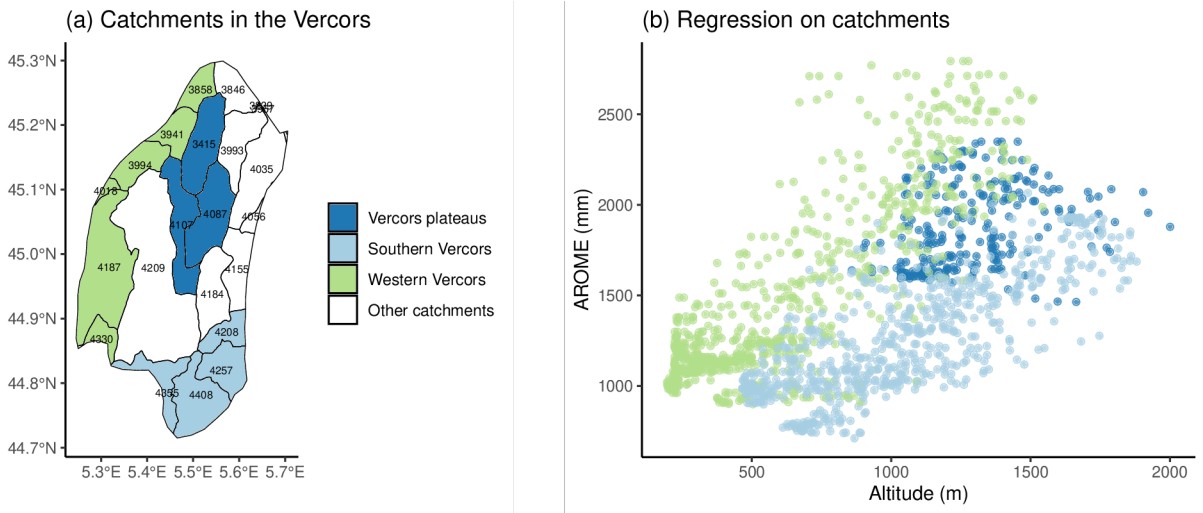

**Figure 8. (a)** Catchments located in the Vercors massif and colored according to their geographic locations. Three groups of catchments are distinguished: the Vercors plateaus, the Western Vercors, and the Southern Vercors. ID numbers are printed for each catchment. **(b)** Scatter plot of AROME annual precipitation and altitude for the pixels in the Vercors (gray dots). The colors of dots correspond to the colors used in panel a).

In the French Alps, a complex topographical region, it is not appropriate to calculate PLRs on too vast sub-regions with surface areas of more than 1,000 km$^2$. In vast sub-regions, there is a risk of mixing areas with too much spatial heterogeneity and losing information about PLR spatial variability. Investigation of PLR with stations or stations-based precipitation products such as SPAZM should be conducted carefully in sparse station networks. For example, in the Aravis massif covering 1,070 km$^2$, linear regressions between precipitation and altitude in SPAZM are performed using all stations within the massif to ensure robustness. However, Fig. 6**(d)** indicates that the Aravis has high intra-massif variability of winter PLR with lower values in its western part. SPAZM can not capture this variability.



## 5.4  Linearity of the relationship between annual precipitation and altitude

In our formulation (Eq. 6), PLRs are defined as the slopes of the linear regressions. The linearity of the relationship between annual precipitation and altitude at the catchment scale is a questionable hypothesis. For example, Avanzi et al. (2021) found non-linearity in two catchments located in the Italian Alps. Smoothing splines provide more flexibility than linear regressions to
model annual precipitation through altitude. However, smoothing splines are not interpretable in terms of PLRs. For this reason, we use piecewise linear regressions implemented in the ℝ package *segmented* (Muggeo, 2008). We perform piecewise linear regressions (unknown break-point between the first and third quartile of altitude) between annual precipitation and altitude on all catchments with sufficient variability (standard deviation of the altitude higher than 50 m) in altitude. To compare the linear ($M_0$ model) and the piecewise ($M_1$ model) linear models, taking inspiration from the Bayes factor defined in the Bayesian
paradigm (Kass and Raftery, 1995), we compute the logarithm of the BIC ratio ($B_r$) as:

$$B_r = \frac{BIC_{M_0} - BIC_{M_1}}{2} \tag{7}$$

If the logarithm of $B_r$ equals zero, the two models fit the data with the same quality. Negative values indicate better fits and positive values refer to worse fits than the linear regression.

| $B_r$ category (log scale) | Interpretation |
|---|---|
| <0 | Anecdotal evidence for $M_0$ |
| 0 to 4.5 | Anecdotal evidence for $M_1$ |
| 4.5 to 9 | Moderate evidence for $M_1$ |
| 9 to 13.5 | Strong evidence for $M_1$ |
| >13.5 | Decisive evidence for $M_1$ |

**Table 3.** Interpretation of the chosen categories of $B_r$. $M_0$ refers to the linear model and $M_1$ to the piecewise linear model.

Figure 9(**a**) illustrates the BIC ratio categories (Table 3) with some examples of scatter plots and corresponding regressions.
Colors refer to categories of the logarithm of $B_r$. Based on the scatter plots, we consider the relationship between annual precipitation and altitude not linear when there is strong or decisive evidence for $M_1$ (red color). Figure 9(**b**) shows the spatial distribution of the logarithm of the $B_r$ values. 10 % of the catchments show decisive evidence for $M_1$. It appears mainly in the Massif Central, the Jura, and the Southern Alps. In the other regions (Northern Alps, Pyrénées), it is sufficient to use linear regression to derive PLR. Figure 9(**c**) shows the distribution in surface areas of the catchments among the five categories of
$B_r$. Strong $B_r$ are mainly found for large catchments, revealing the spatial scale dependence of PLR discussed in subsection 5.3. Too much spatial heterogeneity is still present in some catchments (last scatter plot in Fig. 9(**a**)), suggesting an even finer spatial division, taking into account the aspect and the slope of the terrain faces. Large catchments can have slopes with different orientations for instance. Figure S12 in the Supplements provides the same diagnostic of linearity for AROME seasonal precipitations.



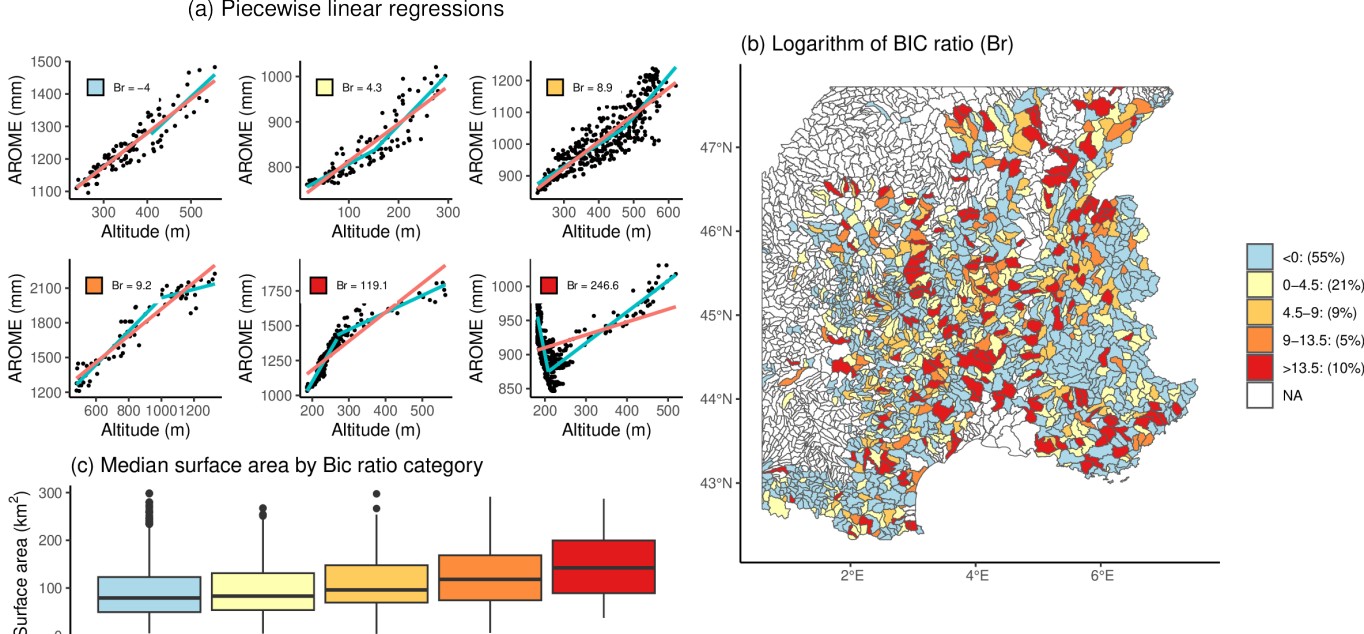

**Figure 9. (a)** Scatter plots of AROME annual precipitation (mm) as a function of the altitude for some catchments taken as illustrative examples of the different categories of BIC ratio. Linear (red lines) and piecewise linear (blue curves) regressions are superimposed. **(b)** Spatial representation of the logarithm of the BIC ratios between linear and piecewise linear regression (annual precipitation ∼ altitude) models. The numbers expressed in % correspond to the percentage of catchments within the classes of the BIC ratio. **(c)** Distribution of the surface area of the catchments according to the categories of BIC ratio.

## 5.5 Uncertainty in the study and outlooks

A limitation of our study is the potential undercatch of precipitation by the high-altitude stations. The underestimation can be as high as 50 % in winter during meteorological events with strong winds and a large proportion of snowfall. A 25 % underestimation of mountain annual precipitation is possible (Sevruk, 1997). In low altitudes, the underestimation is less severe. As a result, the comparison of PLRs in Fig.3**(a)** can lead to wrong conclusions on the bias of the gridded precipitation products. For the same magnitude of bias, our preference goes to seasonal precipitation overestimation rather than underestimation. Despite the measuring uncertainty, we decided to include NPT in our study to account for high ungauged altitude regions. The available period from NPT differs from those of rain gauges and precipitation products. Additional data sources can enhance the spatial and altitudinal representation of PLR. High-altitude measurements of snow water equivalent (SWE) levels can constrain the regressions between winter precipitation and altitude and make PLRs more robust. However, the transition from daily SWE to seasonal precipitations is highly temperature and snowmelt-dependent. For high altitude measurements, it





is possible to make the strong assumption that only solid precipitation has occurred and that snowmelt has been minimal, as stated in Avanzi et al. (2021). Due to the high level of uncertainty of the measure, we opt not to include those data.

At high altitudes, winter precipitation from AROME is significantly higher than that of the other precipitation products, including SPAZM. At 2700–3000 mp in the Pyrénées, AROME simulates an extra 400 mm of precipitation compared to
COMEPHORE, CERRA-Land, and SPAZM. SPAZM may underestimate high-altitude precipitations due to precipitation under-catch. It is also likely that AROME overestimates seasonal precipitations. Indeed, Monteiro et al. (2022) found excessive snow accumulation with AROME in the French Alps above 1800 m. The authors advanced assumptions about spurious snow accumulation, such as the AROME cold temperature bias and the weakness of the AROME snowpack schemes. The winter PLR values given in Fig.7(**b**) for high-altitude catchments should be considered with precaution. In addition to AROME,
various CP-RCMs listed in (Ban et al., 2021, Table1) have been used to simulate precipitation at a fine resolution in France. PLRs derived as the slope of linear regressions are necessarily uncertain because of the regression uncertainty. A perspective of the study could be to include prediction intervals, *i.e.* a range of values containing the "true" value with a probability of 0.95, for example.

## 6  Conclusion

We address the question of the spatial variability of seasonal precipitation lapse rates in France, a country with varied and complex topography. A dense rain gauge network enriched with snow rain gauge totalizators characterizes the relationship between annual/seasonal precipitations and altitude in large regions. The ability to reproduce this link obtained from ground measurements is analyzed using seven different kinds of gridded precipitation products: ERA5-Land, the PDIR satellite product, the raw radar product SERVAL, a 37-year simulation from the CP-RCM AROME, two reanalysis named CERRA-Land
and COMEPHORE, and the precipitation interpolator SPAZM. Precipitation products commonly used in hydrology are limited in high-altitude regions with complex topography, such as the French Alps. Radar products perform well in hilly terrain but underestimate annual precipitation in high altitudes, resulting in a negative association between annual precipitation and altitude. Satellite products are not subject to the same estimation errors as radar products. However, the coarse resolutions do not accurately represent the annual precipitation altitude dependence in regions with topography heterogeneity. The perfor-
mance of station-based products is dependent on the density of the weather station network. In ungauged mountainous areas, precipitation estimation from those kinds of products requires some degree of interpolation, whereas AROME uses physical laws. AROME does not incorporate rain gauge data and is therefore not subjected to bias caused by the under-sampling of particular high-altitude spatial areas. The simulations from the CP-RCM AROME, despite an overall overestimation of high-altitude precipitations, offer an opportunity to assess the altitude dependence of seasonal precipitations at the catchment scale.
We employ the CP-RCM AROME to derive seasonal PLRs on 23 French Alps massifs and 2,748 catchments. The spatial-scale dependence of PLRs is investigated in the French Alps by comparing seasonal PLR values at the sub-regional (1,000 km$^2$) and the catchment (100 km$^2$) scales. PLRs are derived using linear regression between seasonal precipitations and altitude. PLRs are spatial scale-dependent as they tend to be higher and better represented when computed at the catchment scale. The local

topography influence on seasonal precipitations is major at a small spatial scale but dominated by the large-scale atmospheric circulation influence at the larger sub-regional spatial scale. PLRs are higher in winter. Winter PLRs are in the majority positive. 95 % of them are in the range 0.55–13.10 %/100 m. The higher values are found in the Westerly and Mediterranean dominant flow-exposed regions, reaching up to 32 %/100 m. In high-altitude regions such as the French Alps and the Pyrénées, we notice sheltering effects with PLRs close to 0 %/100 m for some catchments enclosed by mountains. It is generally the first reliefs encountered by air masses that host the stronger PLRs. This article emphasizes the importance of considering the topography on a fine spatial scale to estimate PLRs in ungauged mountainous regions. CP-RCM models offer an opportunity to compute the enhancement of seasonal precipitations with the altitude.

**Code and data availability**

SERVAL and COMEPHORE are available from the Aeris portal (https://radarsmf.aeris-data.fr/, last access: 21[th] December 2023). PDIR is available from the CHRS Data Portal (https://chrsdata.eng.uci.edu/, last access: 21[th] December 2023). ERA5-Land is available from Copernicus (https://cds.climate.copernicus.eu, last access: 21[th] December 2023). CERRA-Land is available from Copernicus (https://cds.climate.copernicus.eu, last access: 21[th] December 2023). AROME is available from the Med CORDEX Portal (https://www.medcordex.eu/search/index.php, last access: 21[th] December 2023). The spatial dataset containing the catchments is available from data.gouv (https://www.data.gouv.fr/fr/datasets/, last access: 21[th] December 2023). SPAZM and the stations are not available. The ℝ codes used to perform the analysis are available upon reasonable requests by directly contacting the first author.

**Author contributions**

GE and VD conceived the idea. VD carried out the analysis and wrote the article. GE, DP, and AF participated in the discussion and design of this study and contributed to writing and editing the paper.

**Competing interests**

The authors declare no conflict of interest.

**Acknowledgements**

This work benefits from the exchanges between the researchers of the CNRM (Centre National de Recherches Météorologiques) team of Meteo-France, the researchers of IGE, and the engineers of EDF: special thanks to Martin Ménégoz and Samuel Morin.



**Financial support**

This research has been supported by Electricité de France (EDF).



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
