# Peer review of "Spatial variability of seasonal precipitation lapse rates in complex topographical regions - application in France"

_EGUsphere, 2023_

## Author Comment (AC1)

**Reviewer #1**

**RC1.1 Overall, I believe that the manuscript contains relevant and interesting content that deserves to be published. Furthermore, it is well written and understandable. I recommend publishing it after the authors have kindly considered the few aspects that I would recommend improving.**

We thank the reviewer for this positive feedback.

**RC1.2 The only general aspect I would recommend improving is in the Results and Discussion sections. The results are explained with so many values and details that the reader runs the risk of losing the overview. This is tried to be remedied in the Discussion and Conclusion. Neverthelss, I would try to take better care and add, after or before each section explaining much details, a sentence summarising and explaining the highlights and the general picture. Sometimes the authors have already summarised the results, but if this could be improved a little more I think it would help the reader a lot.**

Thank you for this recommendation. This is a good suggestion and we will add key messages wherever it is necessary.

**RC1.3 Affiliation after the author list: wrong space at the beginning "3 Univ. Grenoble Alpes, CNRS, IRD, Grenoble INP, Grenoble, France"**

Thank you for noticing this wrong space, this will be corrected.

**RC1.4 L104: please reformulate this sentence better introducing "Italy phenomenon"**

Thank you for this comment. We agree that the sentence was unclear. We propose to replace it with the following sentence:

*"In winter, easterly weather fronts coming from the Italian Alps (Garavaglia et al., 2010) can bring large amounts of precipitation in the most Easterly regions of the French Alps.Generally, these meteorological events do not reach the foothills of the French Alps."*

**RC1.5 L120: I got stuck on the word massif, then realised that in the next sentence it is explained and put between apostrophes. Please introduce it better..**

Thank you for this comment. We propose to replace l. 119-122 by:

"In this study, we consider three spatial scales: the regional scale (10,000 km2), the sub-regional scale (1,000 km2), and the catchment scale (100 km2). The sub-regional scale consists of areas named "massif", which forms a set of continuous reliefs, often separated by rivers and valleys. An ensemble of massifs constitutes a mountain range (Alps, Pyrenees). In the French Alps, 23 massifs have been identified through climatological homogeneity of precipitation in Pahaut (1991). PLR computation will be conducted at the massif (climatological homogeneity) and catchment (hydrologic interpretation) scales."

#RC1.6 Already in Fig. 1 it would be interesting to know the area in the different altitude classes as well as the number of stations.

We thank the reviewer for this suggestion. We propose to replace Figure 1-d by the following figure which also shows the surface area for each band of altitudes:

[Figure]

#RC1.7 1: It would be useful to reduce the white space between the subfigures and enlarge them. They are currently very small and difficult to read.

Thank you for this suggestion. We agree with the reviewer that the subfigures are difficult to read. The extra white spaces will be removed from Figure 1.

#RC1.8 L190-191: The other datasets also cannot be defined as homogeneous (change in instrumentation, station density,...)

Thank you for this suggestion. We will add the following sentence: "Some gridded precipitation products present non-homogeneous data because of their large temporal depths. Most radars have been integrated since 2006, and others have been gradually incorporated since 2015 to fill the gaps of measures in mountainous regions (Beck and Bousquet, 2013). The use of SERVAL and COMEPHORE is therefore tainted with temporal non-homogeneity. Station density also affects the temporal homogeneity of COMEPHORE, CERRA-Land, and SPAZM. Change of instrumentation have been corrected in COMEPHORE and SPAZM with an homogenisation of rain gauges precipitation (Gottardi 2009, Mestre et al., 2013)."

#RC1.9 L215-219: What influence does this correction have on the PLRs?

The correction of AROME concerns very few pixels (less than 0.05 %). It impacts very slightly the PLRs, which were computed with a lot of pixels.On the figure below, the corrected pixels are colored in red. The maximum number of corrected pixels is in summer (JJA). They are all located in the Alps.

[Figure]

First correction of AROME

For the summer season (JJA), this correction of AROME mostly modifies the PLRs for two catchments in the Mont-Blanc massif, with an increase of PLRs values. It does not change the interpretation of the spatial variability of PLRs.

[Figure]

PLR (%/100 m) with corrected AROME          PLR (%/100 m) with raw AROME

**RC1.10 3 (b): Hard to see the black line. In addition the abbreviation "RG" is explained in the manuscript, but it would be helpful to mention it again in the Figure caption.**

Thank you very much for this suggestion. We agree with the reviewer that the black line was difficult to discern and that the abbreviation explanation was missing. We propose to replace figure 3 by the following figure.

[Figure]

(a) Annual precipitation by altitudinal bands at stations

(b) Seasonal precipitation by altitudinal bands

**#RC1.11 L403: How was the value of 0.5 chosen?**

This threshold of 0.5 for the $R^2$ was chosen arbitrarily. It does not affect the PLR values, but it helps the interpretation of these values. In our specific case, an $R^2 = 0.5$ suggests that 50 % of the seasonal precipitation variability can be explained by the altitude. 50 % of this variability is significant in our context, given that the variability of precipitation cannot be totally explained by the altitude. This threshold is not uncommon and has been used in other studies (Sevruk, 2002).

**#RC1.12 L549: "m" instead of "mp"**

Thank you for noticing this error. This will be corrected.

---

## Author Comment (AC3)

**Reviewer #3**

**RC3.1 Based on my personal reading, I find the paper interesting and containing detailed and well-conducted analysis.**

We thank the reviewer for this overall positive opinion on this study and his constructives comments.

**RC3.2 With many analyses done (many precipitation products, massif scale, catchment scale), I know you have many results to describe. But I find difficult to not be lost while reading sections 4 and 5, and difficult to get the picture of the main findings. I suggest to, for example, split section 4.1 in two parts (for example, the second focusing just of the 4 products, from line 340), or to clearly summarize the main conclusions at the end of each sub-section (as done for example at lines 369-377).**

Thank you for this recommendation which has also been made by the other reviewers. This is a good suggestion and we will add key messages wherever it is necessary.

**RC3.3 If possible, I suggest to increase the size of some figures and text in figures. It's difficult to distinguish lines or read the text in: Figure 1a, Figure3 (very difficult to distinguish the lines), Figure 6a,b (name of massif and numbers are very small; maybe it could help to have a table listing names, values of R^2 and PLR, and just number in the figure to identify the massif in the list)**

We thank the reviewer for those suggestions. Figure 1 will be larger because we will remove extra white space between the sub-figures (see our response to comment #RC1.7). We also increased the size of the text in figure 1a. Figure 3 is of maximum size. We propose a new version of this figure to better distinguish the reference, which is the black line (see our response to comment #RC1.10).We will add a table listing names, values of R^2 and PLR for figure 6a,b.

**RC3.4 Figure 1: "SAFRAN" name is mentioned in the caption, not in the text. What is it referring to?**

Thank you for noticing this oversight. The 23 massifs are used in the precipitation reanalysis SAFRAN. However, mentioning this reanalysis is useless for the reader's understanding. We propose to remove SAFRAN.

**RC3.5 Figure 4: Is the comparison different in the different regions? could it be possible to distinguish with colors the points for the different regions? Do they cluster? Is there a region more aligned with bisect? The precipitation products are represented by the**

different columns, so maybe colors could be used for giving further information on the comparison in different regions.

We thank the reviewer for this suggestion. We agree that colors show redundant information, already provided by columns.

[Figure]

The subdivision of the points by region does not provide any supplementary information. The points do not cluster. We propose to keep the original figure 4 to conserve the color code by product and the graphical coherence between the figures.

**RC3.6 Figure 7: "insufficient altitudinal variability" should be when "standard deviation is lower than 50 m" (not higher)**

Thank you for noticing this error. It will be corrected.

**RC3.7 Section 5.4. I suggest to move lines 512-523 and table 3 into the methodology section.**

Thank you for this recommendation, l. 512-523 and table 3 will be moved into the methodology section. We also propose to move l. 524-534 and figure 9 into a third result subsection.

**RC3.8 Line 540: it's not clear to me the meaning of this sentence.**

Thank you for detecting this unclear sentence which has been removed.

---

## Author Response (AR1)

**Reviewer #1**

**RC1.1 Overall, I believe that the manuscript contains relevant and interesting content that deserves to be published. Furthermore, it is well written and understandable. I recommend publishing it after the authors have kindly considered the few aspects that I would recommend improving.**

We thank the reviewer for this positive feedback.

**RC1.2 The only general aspect I would recommend improving is in the Results and Discussion sections.  The results are explained with so many values and details that the reader runs the risk of losing the overview. This is tried to be remedied in the Discussion and Conclusion. Neverthelss, I would try to take better care and add, after or before each section explaining much details, a sentence summarising and explaining the highlights and the general picture. Sometimes the authors have already summarised the results, but if this could be improved a little more I think it would help the reader a lot.**

Thank you for this recommendation. We had at the end of all Results and Discussion subsections the following summaries:

L448-452: *To summarise, PLRs are generally higher in winter than in summer. PLRs are sensitive to the spatial scale, with higher values at the catchment scale than the sub-regional (massif) scale. Spatial patterns of significant PLRs are seasonal dependent in France. In winter, the higher PLRs reach 32 %/100 m and are mainly located in the Massif Central. In summer, PLR values are lower and are maximal on the Mediterranean coast. PLRs vary notably in space, even within a single mountain range such as the French Alps.*

L466-468: *In 90 % of the catchments, the regression between seasonal precipitation and altitude can be considered linear. The remaining 10 % often correspond to larger catchments, implying the need to perform linear regression on small climatologically homogeneous catchments.*

L516-520*: To recap, the simulations of the AROME weather model using physical laws offer an opportunity to derive PLRs at a fine spatial scale despite winter precipitation amounts overestimation. The other grid precipitation products emerging from radar, satellite, or rain gauge data show their limits in a complex topography region. Radar is among others subjected to beam blocking. Satellite resolution is too coarse to represent the altitude effect on precipitation. Rain gauge networks are too sparse and even high-density ones do not account for the spatial and altitudinal variability of seasonal precipitation.*

L545-549*: To recapitulate, winter PLRs are higher because of the omnipresence of stratiform events that cause structured orographic precipitation in the first altitude western catchments. In summer, the air masses are coming from the Mediterranean Sea, causing high PLRs in the coastal and hilly Mediterranean areas. In this season,*

*precipitation is mainly caused by the combination of multiple convective events, resulting in unclear PLR spatial patterns.*

L571-573 & L584-585: *"To abridge, PLRs are spatial scale-dependent. The linear regressions give better R2 and larger PLRs at the catchment scale (100km2) than at the sub-regional scale (1,000 km2). Too broad sub-regions often hide distinct altitude effects on precipitation."*

**RC1.3 Affiliation after the author list: wrong space at the beginning "3 Univ. Grenoble Alpes, CNRS, IRD, Grenoble INP, Grenoble, France"**

Thank you for noticing this wrong space, this will be corrected. Meanwhile, the second and third affiliation were merged. We remove the third affiliation.

**RC1.4 L104: please reformulate this sentence better introducing "Italy phenomenon"**

Thank you for this comment. We agree that the sentence was unclear. We replaced it with the following sentence L104-107:

*"In winter, easterly weather fronts coming from the Italian Alps (Garavaglia et al., 2010) can bring large amounts of precipitation in the most Easterly regions of the French Alps. Generally, these meteorological events do not reach the foothills of the French Alps."*

**RC1.5 L120: I got stuck on the word massif, then realised that in the next sentence it is explained and put between apostrophes. Please introduce it better.**

Thank you for this comment. We add L121-126:

*"In this study, we consider three spatial scales: the regional scale (10,000 km2), the sub-regional scale (1,000 km2), and the catchment scale (100 km2). The sub-regional scale consists of areas named "massif", which forms a set of continuous reliefs, often separated by rivers and valleys. An ensemble of massifs constitutes a mountain range (Alps, Pyrenees). In the French Alps, 23 massifs have been identified through climatological homogeneity of precipitation in Pahaut (1991). PLR computation will be conducted at the massif (climatological homogeneity) and catchment (hydrologic interpretation) scales."*

**RC1.6 Already in Fig. 1 it would be interesting to know the area in the different altitude classes as well as the number of stations.**

We thank the reviewer for this suggestion. We replaced Figure 1-d by the following figure which also shows the surface area for each band of altitudes:

[Figure]

(d) Elevation of stations and pixels

**RC1.7 1: It would be useful to reduce the white space between the subfigures and enlarge them. They are currently very small and difficult to read.**

Thank you for this suggestion. We agree with the reviewer that the subfigures are difficult to read. The extra white spaces have been removed from Figure 1.

**RC1.8 L190-191: The other datasets also cannot be defined as homogeneous (change in instrumentation, station density,...)**

Thank you for this suggestion. We removed this sentence at the end of the subsection "COMEPHORE", and added the following sentences at L224:229: *"Some gridded precipitation products present non-homogeneous data because of their large temporal depths. Most radars have been integrated since 2006, and others have been gradually incorporated since 2015 to fill the gaps of measures in mountainous regions (Beck and Bousquet, 2013). The use of SERVAL and COMEPHORE is therefore tainted with temporal non-homogeneity. Station density also affects the temporal homogeneity of COMEPHORE, CERRA-Land, and SPAZM. Change of instrumentation have been corrected in COMEPHORE and SPAZM with an homogenisation of rain gauges precipitation (Gottardi 2009, Mestre et al., 2013)."*

**RC1.9 L215-219: What influence does this correction have on the PLRs?**

The correction of AROME concerns very few pixels (less than 0.05 %). It impacts very slightly the PLRs, which were computed with a lot of pixels.On the figure below, the corrected pixels are colored in red. The maximum number of corrected pixels is in summer (JJA). They are all located in the Alps.

[Figure]

First correction of AROME

For the summer season (JJA), this correction of AROME mostly modifies the PLRs for two catchments in the Mont-Blanc massif, with an increase of PLRs values. It does not change the interpretation of the spatial variability of PLRs.

[Figure]

PLR (%/100 m) with corrected AROME          PLR (%/100 m) with raw AROME

**RC1.10 3 (b): Hard to see the black line. In addition the abbreviation "RG" is explained in the manuscript, but it would be helpful to mention it again in the Figure caption.**

Thank you very much for this suggestion. We agree with the reviewer that the black line was difficult to discern and that the abbreviation explanation was missing. We replaced figure 3 by the following figure.

[Figure]

(a) Annual precipitation by altitudinal bands at stations

[Figure]

(b) Seasonal precipitation by altitudinal bands

**RC1.11 L403: How was the value of 0.5 chosen?**

This threshold of 0.5 for the $R^2$ was chosen arbitrarily. It does not affect the PLR values, but it helps the interpretation of these values. In our specific case, an $R^2 = 0.5$ suggests that 50 % of the seasonal precipitation variability can be explained by the altitude. 50 % of this variability is significant in our context, given that the variability of precipitation cannot be totally explained by the altitude. This threshold is not uncommon and has been used in other studies (Sevruk, 2002).

**RC1.12 L549: "m" instead of "mp"**

Thank you for noticing this error. This has been corrected.

**Reviewer #2**

**RC2.1 This is an interesting, well-written and well-thought manuscript.**

We thank the reviewer for this overall positive opinion on this study and his constructives comments.

**RC2.2 All or some of the compared precipitation products include altitudinal effects or corrections for such effects. does this create precipitation lapse rates artificially? A comment on this would be welcome.**

The interpolator SPAZM applies local linear regressions between altitude and precipitation, conditioned by weather types. This approach aims at adding a physical constraint, with the idea that precipitation lapse rates vary through the territory, depending on the type of precipitation event. Most of the time, the linear regressions reflect a local altitudinal effect and the PLRs are not artificial, that is why SPAZM has been largely used in hydrological modeling (Rouhier 2017). However, we acknowledge that poor regression fits (low $R^2$ values) can create uncertain altitudinal effects in some situations.

Concerning AROME, the first pre-treatment does not rely on altitude. It aims at correcting the precipitation at problematic pixels, often in high-altitude areas, and to remove artificial precipitation lapse rates. The altitudinal effect on the other gridded precipitation products is not artificial but comes directly from physical models or spatialized observations (radar, satellite).

**RC2.3 In eq. 6: the conversion from beta to PLR is unclear to me. A slope is bounded between 0 and infinity, so how can it be converted to a bounded interval between 0 and 100? Also, the denominator P^bar is not defined. Is it the average or median precipitation, and over which area?.**

Thank you for this comment. Indeed, we did not indicate that P^bar corresponds to the mean precipitation over the area where the linear regression is performed (either a massif or a catchment). We had the following explanation L282-283.: "*The slope of the regression β is multiplied by 100 twice, first to obtain the result as a percentage and once more to express PLR in percentage per hundred meters (%/100 m), relative to the average of seasonal AROME precipitation (P̄) over the area where the linear regression is performed.*"

The slopes of the linear regressions are not bounded and take values between minus and plus infinity. The altitudinal effect can be negative for hourly/daily precipitation (Formetta 2021). PLRs are not bounded. PLRs are expressed as the increase or decrease of % of the mean precipitation ( P^bar) by 100 m. The figure below illustrates these bounds with simulated and nonrealistic data.

[Figure]

**RC2.4 389-390 and other places: I can see how there can be more variability in the PLR when one focuses on smaller scales, but not how the average R2 can change depending on scale. I may be wrong, but in principle the R2 value of a large area should be the average of its component subareas. Showing numerical values might clear any doubt.**

The R2 value of a large area does not necessarily correspond to the average of its component subareas. The following figure illustrates an example.

[Figure]

The R2 calculated on the entire area is 0.14.  The R2 values calculated on the three sub-areas are 0.93, 0.05, and 0.84. The mean R^2 is (0.93 + 0.05 + 0.84) /3 = 1.82/3 = 0.607 > 0.14. This often happens when several clouds of dots are present.

**RC2.5 Results in figure 7 (and maybe others): These results are only based on AROME, which is not forced on rain gauge data. While AROME does not perform badly in general, it is not perfect either, and is significantly different than other data-based products. Therefore, I am uncertain of the value of conclusions based only on AROME. Would it be possible to draw a similar figure as figure 7 based only on stations data, even if it means that many more catchments will have NA values?**

For most of the catchments, we do not have enough rain gauges to derive PLR using rain gauges (see figure 1c), resulting in a lot of NA values. Furthermore, the PLR values obtained for some catchments where there are only a few gauges may be very uncertain. The figure below these PLR values, and there is no clear spatial patterns.

[Figure]

**RC2.6 The fact that AROME does not use rain gauges is mentioned as an advantage (e.g. l. 461-462), but this poses the following question: if AROME poorly matches the high-elevation gauges, how to know whether it is because of undercatch or because of some deficiency in AROME? If the comparisons were done against stations data that are corrected for a possible undercatch, would AROME still perform as well? I am not asking to do such an exercise, but it could be discussed.**

Thank you for this suggestion. We agree that this point should be discussed. We added the following paragraph L508:515: *"AROME poorly matches the high-elevation gauges, which could be the combined effect of precipitation undercatch (groisman_accuracy_1994,sevruk_regional_1997, pollock_quantifying_2018} and some deficiencies in AROME (Monteiro, 2022). In Figure 3b, AROME shows a strong agreement with the rain gauges in summer, even the high altitude ones. The differences are larger in winter, where precipitations mostly fall as snow. Precipitation undercatch is limited in summer and more important in winter with solid precipitation. Precipitation undercatch could partly explain the differences among AROME and rain gauges."*

**RC2.7 Many of the results are given as figures, but in general it would be good to have tables with actual numerical values (for example in figure 3b the RMSE between stations and models, and also in figure 4)**

We decided not to include numerical values for figure 3b and figure 4. For figure 3b, the Pearson correlation coefficient can be calculated for each region and are shown in the tables below. However, the number of altitudinal bands per region is small (five or six observations depending on the region) and limits the statistical interpretation of these

values. Moreover, the coefficients obtained from the reanalysis that assimilate rain gauges are close to 1, as expected. We fear that these high values, especially compared to those obtained with AROME, could lead to misinterpretations.

Figure 3b winter

| | Foothills Alps | Northern Alps | Pyrenees | Southern Alps |
|---|---|---|---|---|
| AROME | 0.91 | 0.13 | 0.97 | 0.85 |
| CERRA-Land | 0.98 | 0.94 | 0.97 | 0.51 |
| COMEPHORE | 0.99 | 0.98 | 0.84 | 0.98 |
| SPAZM | 0.94 | 0.99 | 0.87 | 0.99 |

Figure 3b summer

| | Foothills Alps | Northern Alps | Pyrenees | Southern Alps |
|---|---|---|---|---|
| AROME | 0.99 | 0.78 | 0.39 | -0.06 |
| CERRA-Land | 0.88 | 0.59 | -0.48 | 0.69 |
| COMEPHORE | 1 | 0.98 | 0.82 | 0.97 |
| SPAZM | 0.99 | 0.96 | 0.99 | 0.95 |

Figure 4

| | DJF | JJA |
|---|---|---|
| AROME | 0.55 | 0.41 |
| CERRA-Land | 0.59 | 0.42 |
| COMEPHORE | 0.85 | 0.56 |
| SPAZM | 0.80 | 0.71 |

**RC2.8 l.61: word missing. higher reliability?**

Thank you for pointing out this unclear wording. "higher" has be replaced by "larger".

**RC2.9 l.180: incomplete sentence, reformulate.**

Thank you for noticing this unclear sentence. We changed it with the following sentence: "Ground echoes are removed using pixel precipitation probabilities, which are derived from a cloud classification filter."

**RC2.10 Figure 2: large areas seem to have a mean precipitation close to 0 (e.g. in SERVAL). is this a feature of the data or an artifact of the color scale?**

Thank you very much for this comment. We have modified this figure using a discrete color scale (instead of a continuous color gradient):

[Figure]

It was both an artifact of the color scale and a feature of the data. In a large part of the Alps, SERVAL is subject to beam blocking, resulting in a large area of suspicious 50-100 mm of annual precipitation.

**RC2.11 l.275: 50mm is a very small altitude difference. should it be 50m?**

Thank you for noticing this typo. It has been corrected by 50 m.

**RC2.12 Figure 4: would be nice to also have numerical correlation values rather than only visual scatters**

See our response to the comment #RC2.7.

**RC2.13 l.477: resume -> summarize**

Thank you for noticing this wrong choice of word, it has been corrected.

**RC2.14 section 5.4: New methods and results should not be introduced in the discussion section. I recommend moving the part on non-linear results higher in the manuscript.**

Thank you for this recommendation. Section 5.4 has been split into two subsections (3.3 method + 4.3 result).

**RC2.15 Figure 9: This figure is very interesting. It shows that larger catchments have more non-linear PLRs. The question that begs for an answer is whether there exists a**

small subdivision of the catchments where all relations are linear, or if some portions of catchments have irremediably non-linear PLRs.

Thank you for this question. Orographic precipitations are influenced by the leeward and windward mountain faces. To optimize the linearity of the linear regression, we should perform it on small catchments for each slope aspect and wind exposition in order to reduce spatial heterogeneity. However, performing the subdivision would likely require a finer spatial resolution. In addition,some catchments clearly exhibit nonlinear PLRs, irrespective of their size (Jiang 2023) (correlation of precipitation and altitude increase with a decrease of the spatial scale, until a given spatial scale where no improvement is noticed).

**Reviewer #3**

**RC3.1 Based on my personal reading, I find the paper interesting and containing detailed and well-conducted analysis.**

We thank the reviewer for this overall positive opinion on this study and his constructives comments.

**RC3.2 With many analyses done (many precipitation products, massif scale, catchment scale), I know you have many results to describe. But I find difficult to not be lost while reading sections 4 and 5, and difficult to get the picture of the main findings. I suggest to, for example, split section 4.1 in two parts (for example, the second focusing just of the 4 products, from line 340), or to clearly summarize the main conclusions at the end of each sub-section (as done for example at lines 369-377).**

Thank you for this recommendation which has also been made by the other reviewers. See our response to comment #RC 1.2.

**RC3.3 If possible, I suggest to increase the size of some figures and text in figures. It's difficult to distinguish lines or read the text in: Figure 1a, Figure3 (very difficult to distinguish the lines), Figure 6a,b (name of massif and numbers are very small; maybe it could help to have a table listing names, values of R^2 and PLR, and just number in the figure to identify the massif in the list)**

We thank the reviewer for those suggestions. Figure 1 will be larger because we removed extra white space between the sub-figures (see our response to comment #RC1.7). We also increased the size of the text in figure 1a. Figure 3 is of maximum size. We propose a new version of this figure to better distinguish the reference, which is the black line (see our response to comment #RC1.10).We added as suggested a table listing names, values of R^2 and PLR for figure 6a,b.

**RC3.4 Figure 1: "SAFRAN" name is mentioned in the caption, not in the text. What is it referring to?**

Thank you for noticing this oversight. The 23 massifs are used in the precipitation reanalysis SAFRAN. However, mentioning this reanalysis is useless for the reader's understanding. We removed the word "SAFRAN".

**RC3.5 Figure 4: Is the comparison different in the different regions? could it be possible to distinguish with colors the points for the different regions? Do they cluster? Is there a region more aligned with bisect? The precipitation products are represented by the**

different columns, so maybe colors could be used for giving further information on the comparison in different regions.

We thank the reviewer for this suggestion. We agree that colors show redundant information, already provided by columns.

[Figure]

The subdivision of the points by region does not provide any supplementary information. The points do not cluster. We propose to keep the original figure 4 to conserve the color code by product and the graphical coherence between the figures.

**RC3.6 Figure 7: "insufficient altitudinal variability" should be when "standard deviation is lower than 50 m" (not higher)**

Thank you for noticing this error. It has been corrected.

**RC3.7 Section 5.4. I suggest to move lines 512-523 and table 3 into the methodology section.**

Thank you for this recommendation, l. 512-523 and table 3 have been moved into the methodology section. We moved l. 524-534 and figure 9 into a third result subsection.

**RC3.8 Line 540: it's not clear to me the meaning of this sentence.**

Thank you for detecting this unclear sentence which has been removed.

---

## Referee Report (RR1)

The authors adapted their manuscript well and took the comments of the three reviewers seriously and implemented them well. I have no further comments. Thank you very much.